# Black-box and Adaptive Dataset Inference for Large Language Models

## Abstract

Large language models (LLMs) are trained on massive, largely uncurated internet-scale datasets that may contain copyrighted content. This raises a critical question whether a given dataset contributed to a LLM's training. Dataset inference (DI) tackles this challenge by extracting membership signals from a suspect set and aggregating the extracted signals with a statistical test. However, the DI method faces two significant practical limitations. First, it assumes gray-box access to token-level probabilities, whereas most commercial LLM APIs expose only generated text. We address this problem by estimating per-token probabilities from label-only outputs, enabling DI in a fully black-box setting. Second, the DI pipeline relies on fixed hypothesis testing via p-values, which requires committing in advance to both a suspect set and a significance level. This is costly for large suspect sets and often inconclusive for small sets, while adding samples adaptively breaks the validity guarantees of p-values. To overcome this limitation, we introduce DI framework based on e-values and sequential testing, where e-values provide anytime-valid guarantees and support optional continuation. This allows evidence to be accumulated iteratively as more suspect samples are queried. Through these two fundamental advances of token probability approximation from label-only outputs and e-value–based sequential testing, we make DI practical for real-world LLM auditing under black-box access with adaptive evidence accumulation.

## 1. Introduction

Large Language Models (LLMs) such as GPT (Achiam et al., 2023), Llama (Dubey et al., 2024; Touvron et al.,

[1]Anonymous Institution, Anonymous City, Anonymous Region, Anonymous Country. Correspondence to: Anonymous Author <anon.email@domain.com>.

Preliminary work. Under review by the International Conference on Machine Learning (ICML). Do not distribute.

2023), Gemini (Team et al., 2024), and Claude (Anthropic, 2023) are trained on massive, uncurated datasets from the internet. These datasets often contain large amounts of copyrighted texts, raising concerns about intellectual property infringement. The risk is not only hypothetical, as recent work has demonstrated that models such as Llama 3.1 can reproduce entire copyrighted books verbatim (Cooper et al., 2025). But the problem extends beyond books: Currently, there are more than 60 active lawsuits (Maini et al., 2025; Sinai) on the unauthorized use of copyrighted texts in LLM training, such as news articles (The New York Times v. Microsoft and OpenAI., 2023) or source code (J. Doe 1 et al. v. GitHub, Inc., Microsoft Corporation, and OpenAI, Inc., 2024; J. Doe 1 et al. v. GitHub., Microsoft, and OpenAI et al., 2024). These developments underscore the urgent need for technical methods that can reliably determine which data was used to train a given LLM.

LLM Dataset Inference (DI) (Maini et al., 2024) has recently emerged as a promising approach for determining whether a suspect dataset was used to train a particular LLM. DI aggregates training membership features across data points in the suspect dataset and applies statistical hypothesis testing to decide if the dataset contributed to model training. Despite its potential, the DI method faces two core limitations in practical settings. First, DI requires *gray-box access* to internal model details to extract membership features, such as full logit vectors or per-token probabilities, which is not available for most state-of-the-art generative models that are deployed via APIs (Achiam et al., 2023; Team et al., 2024; Anthropic, 2023), where only black-box access is allowed. Second, DI relies on *p-value-based statistical testing*, which requires specifying a fixed suspect dataset and significance level in advance. Large suspect datasets increase computational cost, especially in black-box settings where membership feature extraction is more expensive, while small datasets risk inconclusive results. In principle, adding more data points could improve the testing power, but doing so invalidates the statistical guarantees associated with p-values.

To turn DI into a practical tool for identifying LLMs' training data, we propose our novel **B**lack-box and **A**daptive **D**ataset **I**nference (BADI). We present an overview of our method in Figure 1. BADI enables extraction of membership features even under restricted black-box access by approximating per-token probabilities for the target LLM us-

*Figure 1.* **Overview of BADI.** ❶ Sample pairs of data points from suspect and held-out sets one by one. ❷ Compute black-box membership features using tokens from the target LLM. ❸ Use scores from all previously obtained samples to train a scoring model that distinguishes suspect and held-out data points. ❹ Bet on the scoring model's ability to distinguish suspect and held-out sets in incoming data. ❺ Wealth accumulated in the betting process is a direct measure of evidence against the null hypothesis that the target model did not see the suspect data during training. ❻ Add more data points if a desired significance level is not met. ❼ Stop the protocol at any time (A) if a desired significance threshold is crossed and declare the suspect set a *member set*, or (B) if resources are exhausted. In case (B), the evidence may be interpreted as not significant, and the suspect set is declared a *non-member set*, i.e., not used to train the target model.

ing only token similarities. With this approach, we are able to efficiently transform previous gray-box membership features (Carlini et al., 2021; Shi et al., 2024; Mattern et al., 2023; Mitchell et al., 2023) into their black-box counterparts. We also propose novel features for our DI setting, tailored for our black-box probability estimation. Additionally, we equip BADI with a novel testing framework based on e-values (Vovk & Wang, 2021) rather than p-values, where an e-value is a nonnegative *betting score* whose expectation is at most 1 under the null (large e-values instead of small p-values indicate strong evidence). E-values are particularly well-suited for our BADI, as they support sequential, sample-wise analyses with rigorous statistical guarantees, allowing *inference to be terminated at any point without invalidating the resulting guarantees*. Practically, this means we can begin with a small suspect set, add samples as needed, and stop testing once sufficient evidence has been obtained. We validate our method through comprehensive empirical evaluation on closed and open LLMs with different model sizes and testing on diverse data subsets across multiple domains. In summary, our method enables practical and robust auditing for training data of LLMs.

Overall, our four main contributions are as follows:

1. We propose BADI, a novel practical DI method for LLMs that operates with only black-box access to the models and provides anytime-valid statistical guarantees.

2. We design novel black-box membership features and translate existing gray-box features to the black-box setup by efficiently estimating per-token probabilities.

3. We equip BADI with a novel statistical testing framework based on e-values, which supports sequential testing, enabling flexible, sample and compute-efficient inference.

4. We validate our method through comprehensive empirical evaluation on open and closed LLMs with different sizes, on diverse datasets, and across multiple domains.

## 2. Background and Related Work

**Membership Inference (MI).** The goal of membership inference is to detect if a given data point $x$ was used to train a model (Shokri et al., 2017; Carlini et al., 2022). Multiple MI methods have been proposed for LLMs. They mainly operate under ***gray-box settings*** and require access to a model's loss or perplexity. Carlini et al. (2021) leverage the observation that the ratio of a model's perplexity on a training sample to the sample's entropy (computed with the zlib library) or to its perplexity under a reference model is typically lower than for unseen texts. Shi et al. (2024) proposed MIN-K% PROB based on a simple hypothesis that an unseen text segment is likely to contain K% outlier tokens with low predicted probabilities, while an example seen during training of the model is less likely to contain such outliers. Perturbation-based MI techniques (Mattern et al., 2023) compare a model's perplexity on the original $x$ versus its perturbed version $\tilde{x}$. The perturbations may include synonym substitutions, random deletions, character-level typos, or replacements generated by an external language model (Mitchell et al., 2023). The reasoning is that training set members are more robust to such perturbations than non-members.

More recently, ***black-box***, label-only MI have been put forward. They usually approximate output probabilities using a surrogate model: DPDLLM fine-tunes a surrogate on the suspect text and uses text completion task (Zhou et al., 2024), while PETAL estimates the probabilities token by token on a standard GPT-2-XL model as a surrogate (He et al., 2025). However, most existing MI methods for LLMs do not outperform random guessing (Maini et al., 2024; Duan et al., 2024), including even the strong gray-box MIs. Prior studies have attributed their apparent success to distributional shifts between member and non-member data during evaluation (Maini et al., 2024; Duan et al., 2024). This significantly limits the success of MI for identifying training data in LLMs.

**Dataset Inference (DI).** To overcome the limitation of MI-based methods for training data detection, dataset inference for supervised (Maini et al., 2021) and self-supervised learning (Dziedzic et al., 2022) was proposed. In generative models, its goal is to determine whether a given suspect dataset was used to train a model (Maini et al., 2024; Dubiński et al., 2025; Kowalczuk et al., 2025). Therefore, it extracts diverse MI-features for the individual data points, aggregates them,

and applies statistical testing to reliably determine whether the suspect set was used to train the model. However, the DI for generative models faces severe limitations in practical applicability. First, it relies on gray-box MI features whereas state-of-the-art LLMs like GPT, Gemini, or Claude are deployed behind APIs that provide only black-box access. Second, they rely on statistical testing with p-values which is limited to a fixed size of the suspect set and pre-defined significance level. This causes problems because it is not possible to add more data points to the suspect set when the initial test fails without invalidating the statistical guarantees. We overcome these limitations and make DI practical for the state-of-the-art LLMs by extracting black-box features and performing e-value based statistical testing which provides *anytime-valid* guarantees.

**E-values.** An *e-value* (Vovk & Wang, 2021) is a nonnegative random variable $E$ such that for a given null hypothesis $H_0$ with a set of distributions $\mathcal{P}$, every distribution $P \in \mathcal{P}$ follows $H_0 : \forall P \in \mathcal{P} : \boldsymbol{E}_P[E] \leq 1$. Larger $E$ provide evidence against the null hypothesis, and a level-$\alpha$ test rejects $H_0$ if $E \geq 1/\alpha$. Consider randomly ordered data points $Z_1, \ldots, Z_n \sim P$. At step $i \in [n]$, based on the information available up to time $i$, we place a bet against the null hypothesis and obtain an associated e-value. An *e-process* is a sequence $E_{t \geq 1}$ with the property that for *any stopping time* $\tau$, $E_\tau$ remains a *valid* e-value, *i.e.,* $\forall P \in \mathcal{P} : \boldsymbol{E}_P[E_\tau] \leq 1$. This *optional-stopping* property of e-processes lets us test the hypothesis sequentially without fixing the number of suspect data points $n$ in advance. We may stop and reject the null hypothesis as soon as $E_t \geq 1/\alpha$, or continue if we can collect additional suspect data points and enough resources are available. Moreover, the realized e-value holds evidence regardless of any predefined threshold $1/\alpha$. So we can identify the significance level $\alpha$ post-hoc, and adjust it based on the data (Koning, 2024).

A very natural way of designing sequential tests with e-values is by setting up a betting game and associating statistical evidence with wealth accumulated in the game. This introduces an intuitive interpretation of an e-process as a sequence of bets, *i.e., testing by betting* (Shafer, 2021). BADI relies on this approach to obtain the benefits of e-values while keeping the simple intuition behind the framework. In the next sections, we first highlight our design of black-box MI features for BADI and then provide thorough details on our e-value-based statistical testing framework.

## 3. Black-box Membership Feature Extraction

While pure MI methods were shown unsuccessful to reliably detect whether a data point was used during training of an LLM (Maini et al., 2024), in aggregation, they still can provide reliable DI-features. To enable black-box DI, we: (1) design a series of black-box membership inference features, including *sequence-level black-box features* and *token-level black-box features*, (2) *estimate token probabilities* to compute the features, and finally (3) *aggregate the features in a scoring model* to provide strong evidence on the presence or absence of a suspect dataset in the training data of a given LLM.

**Sequence-level Black-box Features.** We evaluate the LLM's ability to complete a given sequence by measuring the similarity between its generated continuation and the true suffix. In the *sequence-standard* approach, this similarity is an indicator of membership. In the *sequence-perturb* setting, we additionally assess how this similarity changes when the input prefix is slightly altered. Member sequences generally produce higher similarity scores and demonstrate greater sensitivity to such perturbations compared to non-members.

**Token-level Black-box Features.** We estimate per-token membership signals without relying on explicit access to token probabilities from the target LLM. Specifically, we adapt a range of gray-box membership inference features to the black-box setting by substituting black-box-derived probability estimates for those typically obtained in gray-box scenarios. For example, we adopt the MIN-K% PROB (Shi et al., 2024) membership inference, which leverages log likelihood of the $K\%$ least-likely tokens in the text, where their high average suggests that the text was likely included in the pretraining data. Notably, we introduce a new feature **STRIP-K% PROB** tailored for our black-box probability estimation. Our new feature reduces token outlier influence by discarding the most extreme $K\%$ token probabilities (both tails), yielding a stronger signal than MIN-K% PROB. We provide further details on all of the per-token black-box features that we use in the Section A.2, including perturbation based approaches (Mattern et al., 2023; Mitchell et al., 2023), reference-model and zlib-compression-based features (Carlini et al., 2021).

**Estimating Token Probabilities.** To estimate token probabilities under black-box access, PETAL (He et al., 2025) leverages a surrogate model $M_s$ that provides per-token probabilities. Given a prefix $p = x_{t-1}, \ldots, x_1$ and ground-truth next token $x_t$, $M_s$ generates $\hat{x}_t$. PETAL then computes the semantic similarity $sim(x_t, \hat{x}_t)$ between the surrogate model's response and the true token. These similarity scores are monotonically mapped to token probabilities $P(\hat{x}_t)$ via a learned function $f$, which is then applied to the target model $M_t$. The original PETAL method utilizes linear regression to construct the mapping function $f$. For $M_t$, given $p$, it generates the next token $\hat{x}'_t$ and its estimated probability is $\tilde{P}(\hat{x}'_t) = f(sim(x_t, \hat{x}'_t))$. The main limitation of *PETAL* is that it relies on the surrogate model $M_s$ to create the mapping function $f$. This introduces significant compute overhead and, as we demonstrate in our empirical

evaluation in Section A.7, is less reliable for many datasets.

Instead, we introduce a novel ***sigmoid-based token probability estimation*** method that estimates token probability without relying on a surrogate model. Specifically, we obtain per-token embeddings using a sentence embedding encoder. For each next token, we compute the semantic similarity (using the dot product) between the embedding of the predicted next token from the target model and that of the corresponding ground truth token. This similarity score is then transformed into a probability using the sigmoid-based function:

$$\tilde{P}(\hat{x}'_t) = \sigma\left(\text{sim}(x_t, \hat{x}'_t)\right) = \frac{e^{\text{sim}(x_t, \hat{x}'_t)}}{1 + e^{\text{sim}(x_t, \hat{x}'_t)}}. \quad (1)$$

Note that we directly use the ground-truth token $x_t$ and the token $\hat{x}'_t$ predicted by the target model $M_t$, instead of the additional intermediate token $\hat{x}_t$ predicted by a surrogate model $M_s$, eliminating the usage of $M_s$ and enormously simplifying and accelerating the approximation of token probabilities compared to PETAL.

**Scoring Model for Feature Aggregation.** To effectively aggregate the black-box features, we compute the weight of each feature according to its importance and calculate a final membership score based on the weights. Following the gray-box LLM DI (Maini et al., 2024), we obtain the weights for black-box features via a scoring model. Unlike the original gray-box DI approach that employs a static train-test split for the scoring model, we introduce a novel ***online scoring framework*** that processes data points incrementally and adaptively as they come and even request more if necessary. Specifically, we process the combined suspect and held-out sets sequentially, where for each pair of data points $Z_i$, we: (1) train a linear regressor as the scoring model on all previous sample pairs $\{Z_1, Z_2, \ldots, Z_{i-1}\}$ to learn feature importance weights, where suspect samples are labeled 0 and validation samples 1, (2) apply the trained scoring model to predict the membership score for the current $Z_i$, and (3) update our scoring model with the newly observed $Z_i$ for subsequent predictions.

This online formulation addresses the major limitation of prior work that all data points are available upfront for training the scoring model and performing the hypothesis test. Our approach is particularly advantageous for practical scenarios where the data arrive incrementally. We can make decisions and update our knowledge base according to new data points and stop when enough evidence is gathered. We further pre-process the black-box features to provide a more reliable input for the scoring model with two steps: (1) We first normalize all feature values such that different features are at comparable scales, and (2) we replace outliers from top and bottom of the distribution with feature means to prevent skewed correlations. After obtaining the weight for each feature through this online procedure, we produce one aggregated membership score for each data point. Those scores form the input for the statistical testing described in the next section.

## 4. Anytime-Valid Statistical Testing

In DI, the problem is to decide whether the membership features obtained from the scoring model of the suspect dataset $X = \{X_t\}$ differ distributionally from those of a held-out dataset $Y = \{Y_t\}$. Classical DI methods use fixed-sample tests (often a t-test) and report p-values (Maini et al., 2021; Dziedzic et al., 2022; Maini et al., 2024). These approaches suffer from two fundamental drawbacks: (1) they require fixing the suspect set size and significance level *in advance*, and (2) they do not support "*peeking*," *i.e.,* monitoring the evidence from the statistical test sequentially without inflating type-I error or invalidating the guarantees. This severely limits practicality, especially in black-box settings, where the cost of computing membership features is very high, making it prohibitive to use for large suspect sets as in the canonical DI.

To address these limitations, we propose an anytime-valid statistical framework based on sequential two-sample testing (Shekhar & Ramdas, 2023). Evidence is collected incrementally from the samples' membership scores, and testing is stopped as soon as sufficient evidence is gathered. The evidence in our framework is measured via e-values (Vovk & Wang, 2021), which support optional stopping and maintain valid type-I error (false-positive) control even with post-hoc significance claims. Our approach does not require to specify the size of the suspect set or the significance level up front, providing both flexibility and rigorous statistical guarantees. While we introduce this statistical framework alongside our black-box LLM DI features, the framework is universal and can be applied to enhance prior DI approaches across domains and modalities. In the following, we detail the main components of our framework and describe the more technical details in Section C.

**Testing by Betting.** We adopt an *anytime-valid* testing framework based on recent advances on *e-values* (Vovk & Wang, 2021; Shafer, 2021; Shekhar & Ramdas, 2023). An *e-value* is a nonnegative statistic whose expectation under the null hypothesis is at most one. Intuitively, it quantifies evidence against the null: large e-values suggest the null is implausible, while small values are consistent with it. Following the betting framework (Shafer, 2021; Shekhar & Ramdas, 2023), hypothesis testing can be viewed as a *betting game* against the null. At each round $t$, the arbitrator observes a pair of data points $Z_t = (X_t, Y_t)$ and selects a payoff function $S_t$ with null-conditional expectation at most one. The bettor's wealth is updated multiplicatively as

$$W_t = W_{t-1} S_t(Z_t), \quad W_0 = 1. \quad (2)$$

The resulting sequence $\{W_t\}_{t\geq 0}$ is a nonnegative martingale under $H_0$, known as an *e-process* (Ramdas et al., 2020; Ramdas & Wang, 2025).

**Betting Strategies.** A good betting strategy is one that ensures quick growth of evidence under the alternative. This can be quantified by log-optimal principles (Kelly, 1956) and their modern developments (Shafer, 2021; Waudby-Smith & Ramdas, 2024; Grünwald et al., 2024). Motivated by those principles, we focus on strategies that approximately maximize expected log-wealth. Such strategies are conservative in the sense that they never stake all capital and therefore avoid ruin (zero wealth). The resulting wealth process $W_t^\star$ is a (test) martingale under $H_0$ and grows at an exponential rate under fixed alternatives. We adopt the stopping time

$$\tau := \inf\{t \geq 1 : W_t \geq 1/\alpha\},$$

which, by Ville's inequality (Ville, 1939), induces a level-$\alpha$ sequential test.

**Instantiation for DI.** In our adaptation of the general two-sample betting framework of Shekhar & Ramdas (2023) to DI, we use a linearized payoff of the form

$$S_t(Z_t) = 1 + \lambda_t u_t,$$

where $u_t$ is a bounded score quantifying the difference between $X_t$ and $Y_t$ given by a kernel-MMD witness (Gretton et al., 2012). Intuitively, one can think of $u_t$ as the outcome of the betting game that the arbitrator is betting for.

In our case $u_t \in [-1, +1]$, where $u_t$ is close to 1 if the arbitrator correctly distinguishes member and non-member points from the pair $Z_t$, otherwise $u_t$ is close to $-1$. The value $\lambda_t$ is the stake, *i.e.,* how much wealth the arbitrator is willing to bet, chosen adaptively from past data. This part is especially important as, the more outcomes of bets the arbitrator observes, the better they understand the data distribution, and the better of a strategy they can choose.

We use an Online Newton Step (ONS) (Hazan et al., 2007) for staking, ensuring conservative but effective wealth allocation. When the suspect and held-out distributions differ, the expected log-wealth grows linearly with $t$, so wealth increases exponentially and the test is sample-efficient. Under the null, the wealth remains close to 1, preserving Type-I control (of false positives). Please, see Section C.1 for more details.

**Anytime Validity and Post Hoc Interpretability.** Our statistical testing framework has two main advantages over fixed-sample p-value tests: *(1) Anytime validity* means that we can monitor evidence sequentially, stop early if sufficient evidence accumulates, or continue as long as resources allow, without undermining statistical guarantees. This results from the fact that the wealth process is a nonnegative martingale under $H_0$, and as stated above, by Ville's inequality (Ville, 1939), we can stop at any time $\tau$ and reject the null if $W_\tau \geq 1/\alpha$ while still controlling the false positive rate at level $\alpha$. Furthermore, our framework offers *(2) Post Hoc Interpretability* which is useful if we do not want to predefine a significance level, or have to stop testing because we run out of resources. Then, we can still report the e-value as valid evidence. This can also allow for higher significance claims. For example, imagine running a test until obtaining $e = 500$. In this case, we can report a significance of 0.002. In contrast, if we had done a p-valued test, with prespecified threshold of 0.05, but obtained a much lower p-value $p = 0.002$, we could only report the significance level of 0.05, even though it intuitively looks like we have much stronger evidence. Together, these properties make our e-value-based statistical framework a natural fit for black-box DI, where data efficiency and rigorous error control are equally critical.

## 5. Empirical Evaluation

We demonstrate that our proposed BADI method reliably determines whether a given suspect dataset was used in the training of an LLM. We further compare our approach to other methods, showing that it consistently outperforms them. Additionally, we provide an ablation study of our BADI across different model types and sizes as well across diverse datasets and domains.

### 5.1. Experimental Setup

**Models and Datasets.** We evaluate our method on the Pythia models (Biderman et al., 2023), trained on subsets of the Pile dataset (Gao et al., 2020). The Pile provides the training and validation splits for Pythia, which we use to instantiate the suspect and held-out sets. The Pile dataset covers a large spectrum of domains, including academia, medicine, legislation, and code, among others. We also test our method using GPT and Gemini models. Additionally, we show how our method generalizes to other model families, such as Qwen2-7B-Base and Qwen2-7B-Instruct models as well as on OLMo-7B (in Appendix in Section E).

**Baselines.** We evaluate our BADI against two other black-box methods for LLM training data detection. Each of the other methods designs a specific feature to measure the membership signal: (1) **Baseline** calculates the similarities between the ground truth sequences and LLM continuations using RoBERTa scores (it does not use the probability estimations), and (2) **PETAL** estimates the probabilities with next-token semantic similarities and calibrates them with a surrogate model (it only uses a single feature and does not need a scoring model). Our BADI uses the sigmoid-based calibration and online scoring to aggregate our 31 black-

box features. We also provide more details in Section B, including additional results for the **CatShift** method, which computes the similarities of LLM continuations before and after finetuning on the suspect set. However, CatShift occurred to be largely ineffective and we report its results only in Section B.3, focusing here in the main paper on Baseline, PETAL and Gray-Box.

**Scoring Model and Two-sample Testing by Betting.** We adopt an online training protocol for the scoring model and our anytime-valid sequential testing. We present further technical details on the scoring model in Section A.4. At round $t$ of the DI, we first observe a pair of data points $Z_t = (X_t, Y_t)$ that we evaluate the current model on to compute the payoff and update the wealth. Then, we add that data point to the list of the observed data points and calculate the weights using least absolute shrinkage and selection operator (LASSO) with regularization coefficient of 0.01. This procedure is repeated until we pass a desired significance level (*e.g.,* $\alpha = 0.05$ or $\alpha = 0.01$).

We estimate the statistical power using Monte Carlo simulations with $T = 500$ independent trials. In each trial, a length $N_{\max}$ of data stream is formed via a random subset without replacement, the sequential test is run with the fixed threshold $1/\alpha$, and the stopping time is recorded if the threshold is crossed, otherwise the trial is counted as a non-rejection at $N_{\max}$. In all experiments, we use kernel MMD with a degree 2 polynomial kernel as the payoff. We map the raw payoff to $[-1, 1]$ via a $\tanh$ squashing, and employ the ONS (Online Newton Step) as the betting strategy with a stake constraint $\lambda_t \in [-\lambda_{\max}, \lambda_{\max}]$, where $\lambda_{\max} = 0.5$. Larger $\lambda_{\max}$ corresponds to more aggressive staking when a signal is present. To evaluate robustness with respect to data order, all experiments are repeated with 25 different random seeds. We report the average wealth (computed in log-space across seeds) together with log-space confidence intervals.

### 5.2. Effectiveness of Testing by Betting

The main advantage of using e-values for hypothesis testing lies in their *anytime validity*: we may continue collecting data indefinitely without inflating the type-I error rate (false positives) and reject the null hypothesis as soon as the e-value crosses the significance threshold (*e.g.,* $W_t \geq 1/\alpha$, so $W_t \geq 100$ corresponds to $\alpha = 0.01$). When an effect exists, the wealth (or an e-value) grows exponentially, whereas under the null hypothesis $H_0$ it does not systematically increase and remains near its initial value ($\leq 1$), which is further discussed in Section C.

This behavior is illustrated in our main result in Figure 2. Across 12 subsets of the PILE dataset (with additional results in Section D), we observe a clear effect: the wealth for comparing membership features of members versus non-members grows exponentially. Conversely, in a negative-

control setting (non-members vs. non-members), the wealth stays close to one, in line with the theoretical guarantees of sequential testing. Consequently, the observed empirical false positive rate is very low, as we present in Table 3, the sequential test achieves a false positive rate of approximately 1% at the 5% significance level.

In contrast to prior work on DI for LLMs, which required at least 1000 samples for verification (Maini et al., 2024), we show that much fewer samples are required for individual datasets in our case and we do not have to decide on this number upfront. For example, for significance level of 5%, this numbers are approximately $59 \pm 2$ for PhilPapers, $106 \pm 8$ for Stack Exchange, $31 \pm 1$ for Ubuntu IRC, and $213 \pm 17$ for Wikipedia. This adaptability of our method is particularly valuable in scenarios where data is scarce or where computing membership features is computationally expensive, such as in the black-box setting. For the ArXiv, Pile-CC, OpenSubtitles, FreeLaw, and GitHub datasets, we reach the 1% significance threshold less frequently within the available 2000 examples. This indicates the need of incorporating additional data points to strengthen the analysis. This is the task that our method is well-suited to address, as it enables seamless accumulation of further evidence as needed.

### 5.3. Effectiveness of Our Black-box Features

We compare the effectiveness of our BADI with black-box features and sigmoid-based probability estimation against other approaches in Figure 3. PETAL uses the probability estimation with the surrogate model and linear regression, while our method leverages many black-box features aggregated with the scoring model. Additionally, Baseline leverages only the *standard* sequence-level black-box feature, which is used in our method along with *sequence-perturb* features. Our results highlight that given enough data points, our method provides strong enough evidence and successfully predicts the membership for data subsets from the Pile, demonstrating effectiveness of BADI on diverse types of texts. Notably, our method successfully prevents false positives in all cases, which means that we do not falsely accuse the model owner of using copyrighted materials. Compared with PETAL and Baseline on the Pythia-12B model, our approach reaches a given significance level with substantially fewer data points, while requiring much less compute than PETAL (we do not use the surrogate model) and adding a relatively negligible overhead with the scoring model to aggregate many more features than both PETAL and Baseline. For example, on Ubuntu IRC, the average stopping time is 31.2 observations, much fewer than 1000 required for the original LLM DI. As another example, on Books3 the Baseline shows no significant effect within 2000 observations, whereas our method achieves TPR=84% at FPR=1% with an average stopping time of only 788.47 observations.

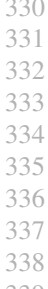 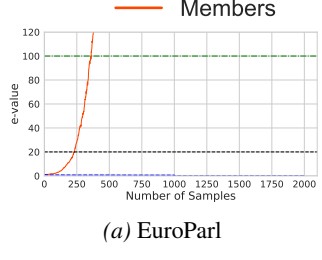 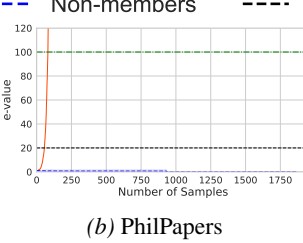 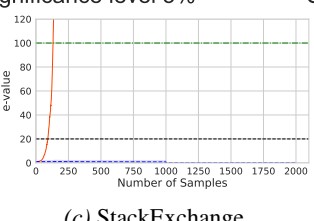 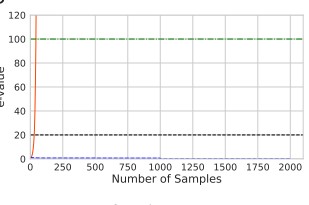

| | | | |
|---|---|---|---|
| *(a)* EuroParl | *(b)* PhilPapers | *(c)* StackExchange | *(d)* Ubuntu |

*Figure 2.* **Testing by betting across Pile subsets.** Accumulated wealth trajectories with 95% confidence intervals (orange shading) for sequential testing by betting on the Pythia-12B model (with more dataset from the Pile presented in Figure 12).

### 5.4. Comparison with Gray-Box DI

We compare the wealth accumulation behavior of BADI under a strictly black-box setting to a gray-box variant in which true per-token log probabilities (or loss values) are available. Our gray-box version of DI corresponds to the original DI for LLMs (Maini et al., 2024), where the brittle p-values are replaced with our anytime-valid statistical testing via e-values. Both settings use identical membership features. The only distinction is that the gray-box variant replaces BADI's sigmoid-based probability calibration with access to the model's actual per-token probabilities. This experiment isolates the effect of probability estimation and directly addresses the question of how much performance is gained by having access to true likelihood information rather than calibrated black-box estimates.

Our results in Figure 3 show that the gray-box setting accumulates evidence against the null hypothesis more rapidly than the black-box setting, confirming that access to exact per-token probabilities provides a statistical advantage. Crucially, however, the performance gap between the gray-box and black-box settings remains modest. This finding demonstrates that BADI's sigmoid-based calibration is highly effective at approximating the information needed for reliable dataset inference.

These results suggest that successful dataset inference does not fundamentally depend on privileged access to internal model statistics. Instead, the key factor is the ability to calibrate membership scores so as to maximize distributional separation between member and non-member samples This is the task for which the sigmoid-based per-token probability estimation is sufficient and robust.

### 5.5. Generalization to Different Model Sizes

We further validate our detection method on three variants of the Pythia models, namely Pythia-410M, Pythia-6.9B, and Pythia-12B. We assess how model scale affects the performance of our method. As shown in Figure 3, our method consistently provides correct dataset identification across different models sizes. We report additional results for more datasets in Section D.4. Overall, scaling up the model sizes

*Table 1.* **BADI performance on black-box LLM APIs.** We apply BADI to state-of-the-art black-box LLMs on the WikiMIA dataset. In addition to area under the ROC curve (AUC), we report true positive rate (TPR), false positive rate (FPR), and average stopping time (mean $\pm$ std), which are obtained via sequential betting-based hypothesis testing.

| Model | AUC | TPR | FPR | Avg. Stop |
|---|---|---|---|---|
| GPT-4o | 0.8359 | 1.0000 | $4 \times 10^{-4}$ | $35.0 \pm 1.3$ |
| Gemini 2.0 Flash | 0.8354 | 1.0000 | $9 \times 10^{-4}$ | $36.1 \pm 2.4$ |
| GPT-3.5-Turbo | 0.7223 | 0.9999 | $2 \times 10^{-4}$ | $61.8 \pm 3.3$ |
| GPT-4 | 0.7369 | 0.9999 | $4 \times 10^{-4}$ | $62.8 \pm 4.7$ |
| GPT-4.1 | 0.7262 | 0.9999 | $2 \times 10^{-4}$ | $73.5 \pm 5.6$ |
| GPT-5 | 0.6159 | 0.7787 | $8 \times 10^{-5}$ | $264.3 \pm 41.6$ |

improves detection on Books3, BookCorpus2, and Gutenberg datasets, with shorter average stopping times. Detection is consistently strong on Ubuntu IRC, StackExchange, PhilPapers, and YouTube Subtitles, with early stopping. However, the scale effects are not uniform: for Wikipedia (and in some cases EuroParl and USPTO), larger models yield a weaker signal.

### 5.6. Generalization To Different Model Families

We further evaluate our DI method in a black-box setting against the commercial LLMs accessed via APIs. We report the results in Figure 4 for the WikiMIA dataset (Fu et al., 2025) as the suspect set, which uses March 2024 as the cutoff date for distinguishing members from non-members. We also observe similar trends in Figure 19 for the the Book-MIA dataset (Shi, 2023), which serves as a benchmark designed to evaluate MIA methods, specifically in detecting pretraining data from OpenAI models with the cutoff year of 2023. As shown in Figure 4 and Table 1, our approach consistently succeeds in this restricted setting: it crosses the 1% significance threshold in fewer than 100 samples for most models, with the exception of GPT-5, and maintains uniform FPR control across 185 observations. On average, our method achieves a TPR of 0.995 with an average stopping time of 125.04 observations. We suspect that the different performance of GPT-5 might stem from it much later release than WikiMIA cutoff date.

Additional evaluation results of BADI across different model families are provided in Section E for the OLMo-

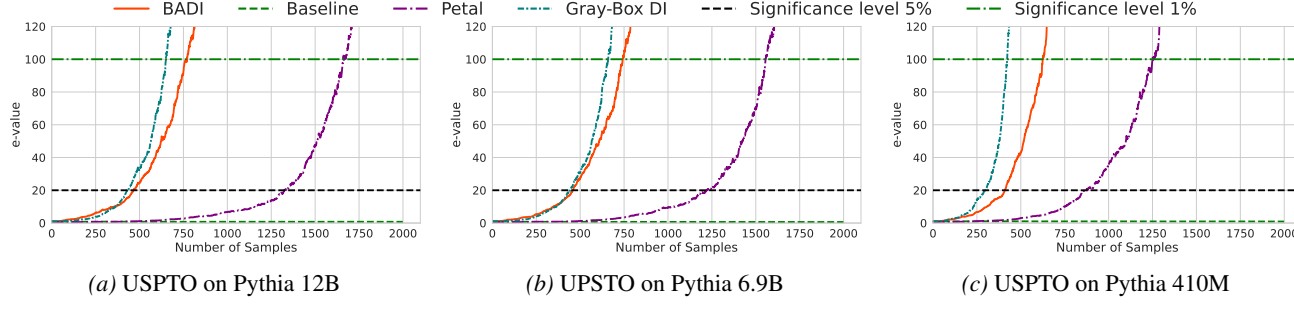

*(a)* USPTO on Pythia 12B     *(b)* UPSTO on Pythia 6.9B     *(c)* USPTO on Pythia 410M

*Figure 3.* **Comparison across methods and model sizes.** We compare BADI against the black-box based PETAL and Baseline methods as well as the Gray-Box DI which is an equivalent method to the original DI but using our e-values (see more datasets in Figure 13).

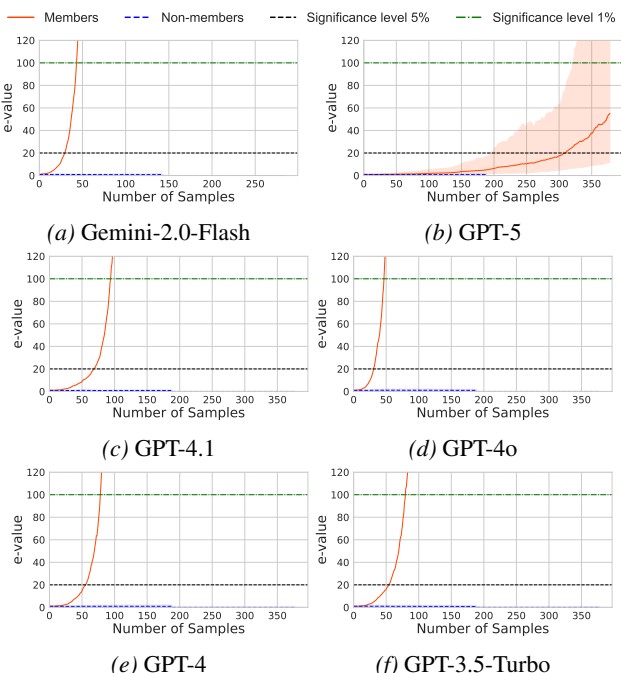

*(a)* Gemini-2.0-Flash     *(b)* GPT-5

*(c)* GPT-4.1     *(d)* GPT-4o

*(e)* GPT-4     *(f)* GPT-3.5-Turbo

*Figure 4.* **BADI for black-box LLM APIs.** We present the wealth trajectory for WikiMIA dataset with Gemini and GPT models.

7B and Qwen2-7B models. These experiments highlight BADI's applicability not only to locally deployed open-weight models, but also to real-world commercial black-box API where internal LLM parameters and logits are not accessible.

### 5.7. Ablation Studies

**Scoring model hyperparameter analysis.** We adopt a regression-based framework as our scoring model and, in particular, use LASSO due to its simplicity and built-in regularization. LASSO provides effective feature aggregation while naturally controlling model complexity. A detailed description of the scoring model is provided in Section A.4, along with an extensive hyperparameter analysis in Section A.5.

**Feature Importance Analysis.** We investigate which black-box features are most significant by training the scoring model on 1000 examples across 20 random data shuffles. We extract the average weight from these shuffles for each dataset. The feature importance weights are shown in Figure 24 (in Section A.6). We observe that all of our 31 black-box features are necessary, including our newly introduced STRIP-K% PROB, and all of the features provide a significant signal for different datasets.

**Token Probability Estimation.** Additionally, we compare three methods for estimating per-token probabilities. Each of them assigns probabilities to the next token $\hat{x}'_t$ from the target model based on the ground truth next token $x_t$ and different transformations of token similarities. The methods are as follows: (1) the raw approach: $1 - sim(x_t, \hat{x}_t)$; (2) the PETAL-based approach: $-sim(x_t, \hat{x}'_t) \cdot \beta - \alpha$, where $\beta$ and $\alpha$ are parameters fitted via linear regression using surrogate model and probabilities; and (3) our sigmoid-based approach: $-\sigma(sim(x_t, \hat{x}_t))$, based on Equation (1), where $\sigma$ denotes the sigmoid function. We show in Figure 7 (in Section A.7) that our sigmoid-based method achieves the highest performance among the three tested approaches.

## 6. Conclusions

We have introduced BADI, a novel black-box and adaptive DI framework for detecting whether a specific dataset was used to train an LLM. Our key innovation is the integration of black-box membership features with sequential and anytime-valid statistical testing, enabling us to perform dataset inference for any LLM using a flexible betting protocol that delivers valid results after any number of evaluated samples. This approach significantly reduces the computational burden of black-box dataset inference by allowing early stopping as soon as sufficient evidence is attained, or continuation for increased statistical evidence as resources permit. Our empirical evaluation, performed on diverse models trained on various datasets, confirms the effectiveness and scalability of our approach. Overall, BADI provides a practical black-box solution for real-world model auditing, supporting individuals' rights to control how their data is used in training of LLMs.

## Impact Statement

The introduction of BADI has significant societal implications, as it empowers users and regulators to audit and verify whether their data was included in the training of LLMs without requiring access to model internals. This capability enhances transparency and accountability in AI development, supports compliance with regulations such as EU AI Act, and advances individuals' rights to data usage consent and control. By reducing the computational barriers to effective dataset inference, BADI is designed to facilitate broader adoption of AI auditing practices, promote more ethical use of data in training LLMs, and foster greater public trust in AI systems.

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

# A. Implementation Details

## A.1. Dataset

We use a dataset from PILE, downloaded from `https://huggingface.co/datasets/pratyushmaini/ll m_dataset_inference/resolve/main`, which is the same dataset as used in the LLM Dataset Inference (Maini et al., 2024). Each subset has ~2000 member samples and ~2000 held-out samples, and each sample has ~512 tokens. For the features based on similarity, we set the maximum length to 512 tokens. For the features based on BERT score, we set the suffix to 64 tokens. For this reason, similarly to previous work, we do not include Enron and NIH data subsets in our evaluation. All samples in the NIH data subset have fewer than 64 tokens and Enron has fewer than 1000 samples.

## A.2. Per-token Black-box Membership Features

We use the following per-token black-box features:

1. **Perplexity.** A basic feature that computes the perplexity or the average loss on the target model.

2. **MIN-K% PROB.** (Shi et al., 2024). Averages log likelihood of the $K\%$ least-likely tokens in the text. A high average suggests that the text was likely included in the pretraining data. In our black-box setting, we find that $K = 0.6$ works best.

3. **STRIP-K% PROB.** *This is our new added feature for the black-box setting.* It reduces outlier influence by discarding the most extreme $K\%$ token probabilities (*e.g.,* both tails) before aggregation (*e.g.,* via averaging), yielding a stronger signal than MIN-K% PROB.

4. **Perturbation-based.** (Mattern et al., 2023; Mitchell et al., 2023) Measures the change in perplexity between the original text and its perturbed variant (*e.g.,* via synonym substitution, random deletion, or paraphrasing by another language model).

5. **Reference-model-based.** Compares the suspect model's perplexity to that of a reference model on the given samples.

6. **Zlib ratio.** (Carlini et al., 2021) Computes the ratio between the model's perplexity and the entropy (number of bits bits) of the text after `zlib` compression (Gailly & Adler, 2004). The underlying intuition is that a model trained on particular data encodes information about it in a more efficient manner than generic compression algorithms such as `zlib`.

## A.3. Text Perturbations

For every prefix, six variants of perturbed text are created using NL-Augmenter `https://github.com/GEM-bench mark/NL-Augmenter`. The perturbations include:

1. **Synonym Substitution.** Words in the text are replaced with their synonyms while keeping the overall meaning intact. In our work we change words with probability 0.25.

2. **Butter Fingers.** Characters are randomly swapped, replaced with neighboring characters to simulate human typing errors. For example for character 'g' it stays unchanged or is swapped with given probability (we use 0.1) with one of following: t, b, f, h, e, d, c, y, j, n.

3. **Random Deletion.** Certain words or characters are randomly removed from the text. This introduces incomplete information to check if the model can still infer context. We delete randomly with probability 0.25.

4. **Change Character Sase.** Letters are randomly capitalized or lowercased (*e.g.,* 'which' → 'WhIch'). Every character can be permuted with predefined probability (we use 0.1).

5. **Whitespace Perturbation.** Extra spaces are inserted or existing ones are removed. This disturbs token boundaries and formatting without altering the words themselves. (e.g. 'because the problem' → 'be ca use thep roblem'). There are separate probabilities for whitespace remove and addition. We use 0.1 and 0.05 respectively.

6. **Underscore Trick.** Spaces are randomly replaced with underscores (e.g. 'Hello world' → 'Hello_world'). Each replacement happens with given probability (we use 0.25).

### A.4. Our Scoring Model

In our experiments, we use a linear regression model as the scoring function, specifically the least absolute shrinkage and selection operator (LASSO). LASSO aggregates the membership features extracted for each data point from the suspect and held-out sets into a single scalar score.

These aggregated scores are then used within a sequential hypothesis testing procedure. At each round $t$, we compute the mean aggregated score for the suspect set and the held-out set using all previously observed samples. A payoff function applied to these mean scores determines the bet placed against the null hypothesis at round $t$. Then, the aforementioned data point is added to the list of observed data points, the LASSO model is re-trained using all accumulated samples. This process continues sequentially until the accumulated evidence crosses a desired significance threshold. Importantly, the significance threshold can be adjusted adaptively over time without inflating the false positive rate, due to the anytime-valid nature of the underlying testing-by-betting framework.

### A.5. Hyperparameter Analysis of the Scoring Model

We use LASSO to aggregate membership features both in our sequential testing framework and in the experiments based on classical p-values. This choice ensures consistency with prior work in the membership inference literature and enables direct comparison with existing methods.

A key advantage of LASSO is its built-in regularization, which promotes sparsity in the learned feature weights. Our underlying hypothesis is that while the full set of membership features used in BADI is informative, only a subset of these features may be relevant for detecting membership within a given data subset. LASSO naturally adapts to this setting by selecting informative features while suppressing irrelevant ones, without sacrificing the simplicity of a linear scoring model.

In all experiments, we use a regularization coefficient of $0.01$. The rationale for this choice is examined empirically by comparing aggregation performance across regularization coefficients of $0.01$, $0.001$, and $0.0001$. For each setting, we run the full BADI pipeline, differing only in the regularization strength used during feature aggregation prior to hypothesis testing. We report the resulting p-values to assess the sensitivity of our method to the choice of regularization.

The results are reported in Figure 5 and show that, despite its simplicity, LASSO remains an effective method for aggregating membership scores. We observe that a regularization coefficient of $0.01$ provides the best balance between true positive and false positive detection. As the regularization strength decreases, the false positive rate is further reduced; however, this reduction comes at the cost of a lower true positive rate. In contrast, a regularization coefficient of $0.01$ achieves strong true positive performance while maintaining a relatively low false positive rate. For this reason, we select $0.01$ as the default regularization value and use it consistently throughout both the static and sequential testing procedures.

### A.6. Feature Importance Analysis

The black-box feature importance weights for member versus held-out data are shown in Figure 24. The figure clearly illustrates the effect of $\ell_1$ regularization in promoting sparsity among the feature weights. This observation supports our initial hypothesis that, while the full set of membership features is collectively informative for dataset identification, not all features are equally relevant for every data subset.

LASSO enables subset-specific feature selection without introducing additional model complexity. In particular, some features receive zero weight for certain subsets while playing a significant role in others. Moreover, the generally small magnitude of the nonzero feature weights indicates that feature aggregation is performed in a controlled manner, providing further evidence that the model avoids overfitting.

### A.7. Methods for Token Probability Estimation

To test all gray-box features, we utilize per-token similarity to emulate the loss. First, for a given prefix $x_1, \ldots, x_{t-1}$ only the next token $\hat{x}_t$ is predicted with "greedy" decoding. To get similarities we create embeddings of the output text and oracle using the embedding model `sentences-transformers/all-MiniLM-L6-v2` and then calculate the cosine similarity per each token. We noticed that similarity differs significantly from log-probability, as shown in Figure 6. Similarity has many values of 1.0 when the model predicts the correct token, approaching $\sim$50% in different datasets.

To address this issue, three distinct methods of estimating the loss function with tokens similarity were tested:

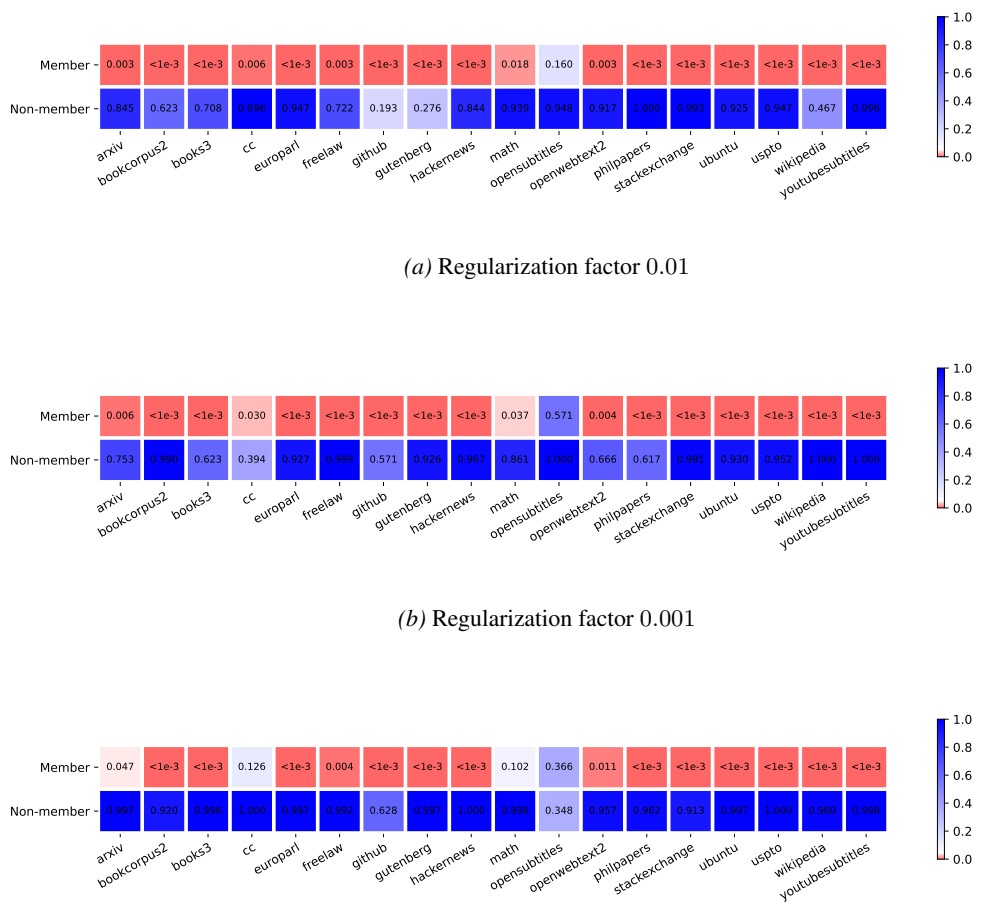

*(a)* Regularization factor 0.01

*(b)* Regularization factor 0.001

*(c)* Regularization factor 0.0001

*Figure 5.* $P$-**value distributions across regularization strengths.** Distributions of BADI performed on the Pythia-12B model across Pile subsets. We observe the sensitivity of BADI performance to the LASSO regularization factor used in the scoring model.

**Raw Similarities.** As tokens are more semantically similar, a cosine similarity goes to 1 and log-loss goes to 0, therefore, the simplest approach was to transform similarity in the following way:

$$-\tilde{p}(\hat{x}_t | x_1, \ldots, x_{t-1}) = 1 - sim(x_t, \hat{x}_t), \tag{3}$$

where:

- $\tilde{p}$ - estimated probability,

- $x_1, \ldots, x_{t-1}$ - prefix,

- $\hat{x}_t$ - predicted next token,

- $x_t$ - ground truth next token,

- $sim$ - similarity function (cosine similarity).

**Reference Model.** This approach follows the original PETAL method. Per-token similarities and oracle token probabilities from the reference model (GPT2-XL) are obtained. Then linear regression is fitted to obtain slope ($\beta$) and intercept ($\alpha$).

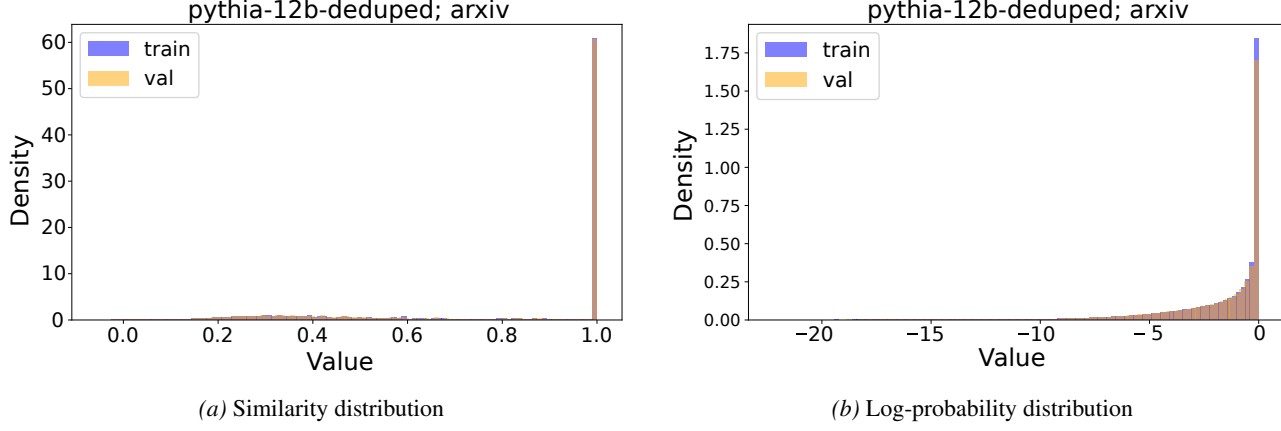

*(a)* Similarity distribution

*(b)* Log-probability distribution

*Figure 6.* **Comparison of Similarity and Log-probability Distributions.** We use the Pythia-12B-deduped model on arXiv dataset for this experiment.

Finally log loss is estimated in following way:

$$\tilde{p}(\hat{x}_t | x_1, \ldots, x_{t-1}) = sim(x_t, \hat{x}_t) \cdot \beta - \alpha \tag{4}$$

**Sigmoid.** Sigmoid function ($\sigma(x)$) takes any real-valued number and maps to range $(0, 1)$. That property makes it an excellent choice to model the probability, then the loss estimation is as follows:

$$\tilde{p}(\hat{x}_t | x_1, \ldots, x_{t-1}) = \sigma(sim(x_t, \hat{x}_t)) \tag{5}$$

The best performing method is our approach that utilizes the sigmoid function. It maps values to a bounded range, which reduces the impact of extreme perplexity outliers and prevents excessive compression during normalization. A comparison of the methods can be seen in Figure 7.

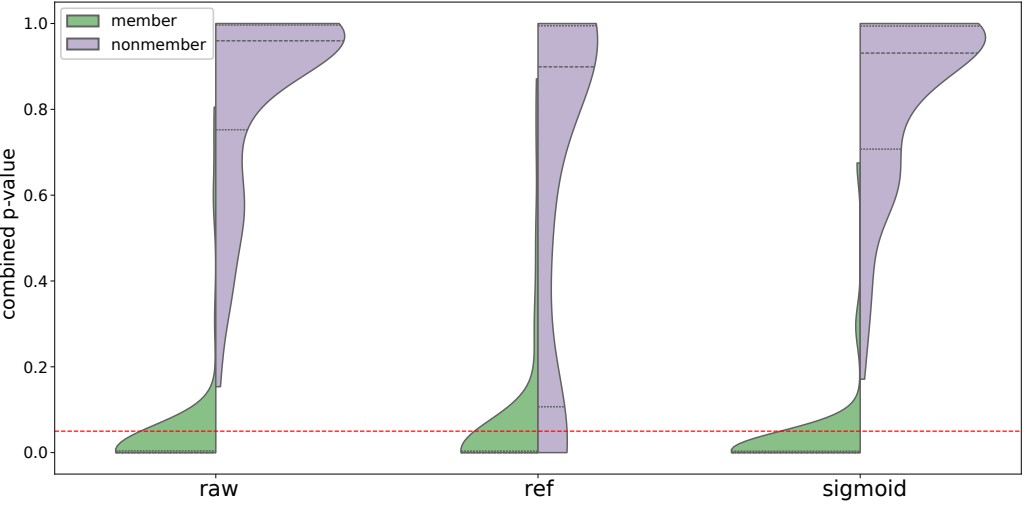

*Figure 7.* **Distribution of p-values Across all Pile Subsets on Pythia-12B-dedup.** For each subset, we run the scoring model 40 times with 1000 samples per run. The resulting p-values are grouped in batches of four and combined within each group. The violin plots display the distribution of these combined p-values aggregated from all subsets for three different methods: raw similarities, reference model (based on PETAL), and our approach based on the sigmoid function. The better method has lower p-values for members (shown in green) and higher p-values for non-members (shown in purple).

### A.8. Normalization and trimming of extreme values

Given a sequence of training metrics $\{t_j\}_{j=1}^n$ and corresponding validation metrics $\{v_j\}_{j=1}^m$, the Z-score normalization using only training statistics proceeds as follows:

Compute the empirical mean and standard deviation of the training metrics:

$$\mu_t = \frac{1}{n} \sum_{j=1}^{n} t_j, \tag{6}$$

$$\sigma_t = \sqrt{\frac{1}{n} \sum_{j=1}^{n} (t_j - \mu_t)^2}. \tag{7}$$

Normalize both training and validation metrics using the training set statistics:

$$\tilde{t}_j = \frac{t_j - \mu_t}{\sigma_t}, \quad j = 1, \dots, n, \tag{8}$$

$$\tilde{v}_k = \frac{v_k - \mu_t}{\sigma_t}, \quad k = 1, \dots, m. \tag{9}$$

Here, $\tilde{t}_j$ and $\tilde{v}_k$ denote the normalized training and validation metric values, respectively.

The normalized training metrics satisfy $\mathbb{E}[\tilde{t}_j] = 0, \quad \mathrm{Var}[\tilde{t}_j] = 1$.

Next, we trim extreme values by removing the 2.5% smallest and 2.5% largest values and replace them with the global mean.

### A.9. Training of the linear model (for p-values only)

Next, we perform the following procedure for multiple random shuffles:

1. Shuffle the training and validation metric vectors independently.

2. Split the shuffled data into a training subset and a held-out subset of equal size.

3. Fit a linear model on the training subset to learn feature weights.

4. Apply the trained model to the held-out subset and conduct a one-sided t-test on the predicted scores to obtain a p-value.

Because the different held-out subsets overlap, the resulting p-values are statistically dependent. To aggregate these p-values $p_1, p_2, \dots, p_n$ into a single combined p-value $p_{\text{combined}}$, we use the Brown–Sidak correction:

$$p_{\text{combined}} = 1 - \exp\left( \sum_{i=1}^{n} \log(1 - p_i) \right).$$

This method is conservative and helps control the Type I error rate when tests are dependent.

### A.10. P-value Tests Across Model Sizes

In order to assess differences across model sizes, we perform one-sided t-tests, from which we derive the corresponding p-values, as presented in Figure 8. These p-values are shown both to enable comparison with e-values and to provide consistency with prior work on dataset inference, where p-values were commonly employed.

## B. Implementation Details for Baseline Methods

We consider three baseline methods for the black-box LLM training data detection. Each baseline reflects a distinct notion of membership signal, quantified through a dedicated approach to measure the different in distribution between the suspect and held-out sets.

We use the following baseline methods:

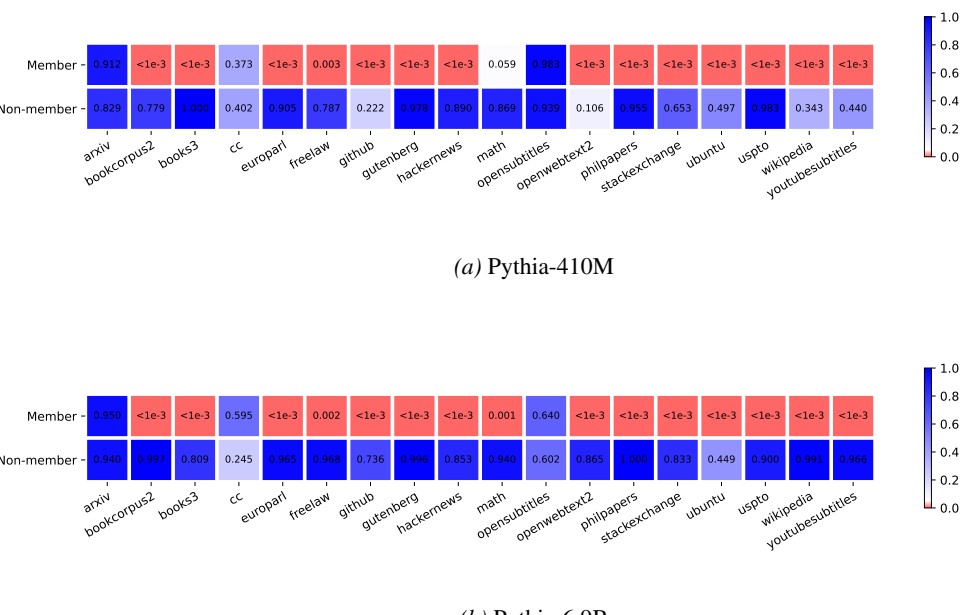

*(a)* Pythia-410M

*(b)* Pythia-6.9B

*Figure 8.* P-value distributions across different Pythia model sizes with regularization factor of 0.01

1. **Baseline.** This is a naïve baseline based memorization of training data by LLMs. This method evaluates the degree to which an LLM reproduces the ground-truth continuations of given sequences. Specifically, model-generated suffixes are compared to the original suffixes using the RoBERTa score, a variant of BERTScore (Zhang et al., 2020) in which the RoBERTa-large model (Liu et al., 2019) is employed. Higher similarity values are interpreted as stronger evidence of memorization.

2. **PETAL.** (He et al., 2025) For each example, PETAL estimates token-level probabilities by measuring the semantic similarity between model-generated and ground-truth tokens using reference model. These similarity scores are used to train linear regression model to map semantic similarities to log-probabilities, which are then aggregated at the sentence level. The resulting values are compared across suspicious and held-out sets.

3. **CatShift.** (Xiong et al., 2025) leverages the phenomenon of catastrophic forgetting, whereby models overwrite previously learned representations during fine-tuning but can also "reactivate" them when re-exposed to familiar data. What CatShift does, it splits a suspicious dataset into a training subset and a testing subset, then fine-tunes a target LLM on the training portion using a text completion task. Model outputs on the testing subset are collected both before and after fine-tuning, and their similarity is quantified using metrics such as BERT-based scores. The distributions of these similarity scores are then compared against those obtained from a known non-member validation set using statistical tests such as the Kolmogorov–Smirnov test. If the suspicious dataset was included in training, fine-tuning induces a stronger output shift, reflecting the reactivation of forgotten knowledge, whereas novel data induce only modest changes.

4. **Gray-Box DI.** Gray-Box DI is a baseline that incorporates the same set of per-token membership features as BADI. The key distinction is that Gray-Box DI directly uses the model's true per-token log-probabilities, rather than relying on calibrated estimates as in the black-box setting. This baseline is designed to quantify the performance gains attributable to access to gray-box per-token probability information. All other implementation details, including feature extraction and downstream inference procedures, are shared between Gray-Box DI and BADI.

### B.1. Baseline: Naïve Approach Based on Memorization

Each test example is split into a *prefix* (448 tokens) and a *suffix* (last 64 tokens). Then, we prompt the target model with *prefix* and generate exactly 64 tokens as the suffix. The generated prediction is compared to the original *suffix* by computing the RoBERTa score (using both the generated suffix and the ground-trugh one). Finally, we perform a one-sided t-test to

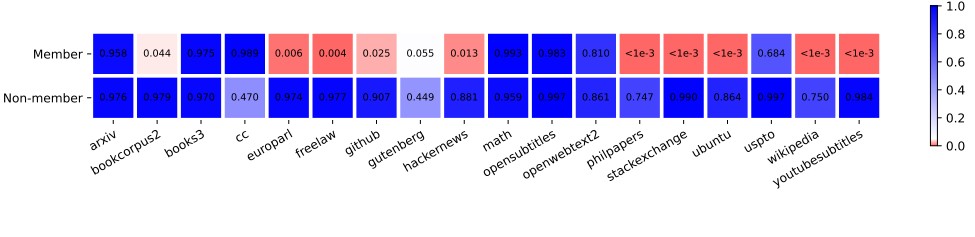

*(a)* Baseline

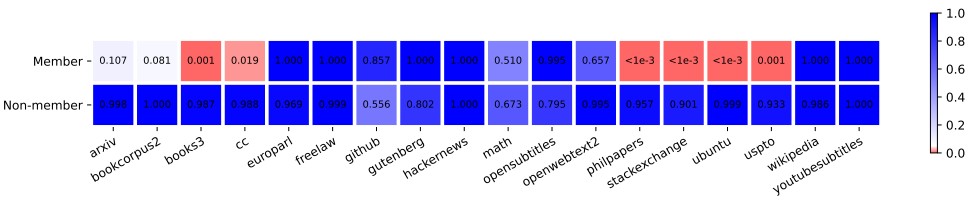

*(b)* PETAL

*Figure 9.* P-value distribution for the (a) Baseline and (b) PETAL methods on Pythia-12B.

assess whether the distribution of RoBERTa scores on the suspicious set differs significantly from that on the held-out set. P-values across data subsets used for this baseline method are presented in Figure 9.

### B.2. PETAL Method

The PETAL method (He et al., 2025) estimates next-token probabilities under black-box access using a surrogate model $M_s$. For each token position $t$, let the prefix be

$$p = x_1, x_2, \ldots, x_{t-1},$$

and let $x_t$ be the ground-truth next token. We query $M_s$ with:

```
--max-new-tokens=1
```

to obtain the surrogate prediction $\hat{x}_t = M_s(p)$ and log-probability $\log P_s(x_t \mid p)$. We then embed both $x_t$ and $\hat{x}_t$ using `sentence-transformers/all-MiniLM-L6-v2`, yielding vectors $\mathbf{e}(x_t)$ and $\mathbf{e}(\hat{x}_t)$. The cosine similarity is computed as

$$\mathrm{sim}(x_t, \hat{x}_t) \;=\; \frac{\mathbf{e}(x_t) \cdot \mathbf{e}(\hat{x}_t)}{\|\mathbf{e}(x_t)\| \|\mathbf{e}(\hat{x}_t)\|}.$$

We fit a linear calibration function $f$ by regressing the surrogate log-probabilities onto these similarities:

$$\log P_s(x_t \mid p) \;=\; \alpha \log\big(\mathrm{sim}(x_t, \hat{x}_t)\big) + \beta,$$

where $\alpha, \beta$ are learned via least squares.

To apply PETAL to the target model $M_t$, we prompt $M_t$ on the same prefix $p$ to generate $\hat{x}'_t = M_t(p)$, compute $\mathrm{sim}(x_t, \hat{x}'_t)$, and estimate its log-probability as

$$\log \tilde{P}_t(x_t \mid p) \;=\; \alpha \log\big(\mathrm{sim}(x_t, \hat{x}'_t)\big) + \beta.$$

For each example in both the suspect set $D_{\mathrm{sus}}$ and the held-out set $D_{\mathrm{hold}}$, we aggregate the estimated log-probabilities:

$$\bar{\ell} \;=\; \frac{1}{T} \sum_{t=1}^{T} \log \tilde{P}_t(x_t \mid x_{<t}).$$

Finally, a one-sided $t$-test is conducted on the distributions of $\bar{\ell}$ over $D_{\text{sus}}$ versus $D_{\text{hold}}$, yielding a $p$-value for each epoch. The resulting distribution of $p$-values is shown in Figure Figure 9.

### B.3. CatShift Method

We use the `sus` dataset comprising 1000 samples, evenly split into a fine-tuning set $D^{\text{train}}$ and an evaluation set $D^{\text{test}}$, each containing 500 samples. The base model $M_{\text{base}}$ is fine-tuned on $D^{\text{train}}$ for 10 epochs to yield $M_{\text{ft}}$. A held-out set $D^{\text{hold}}$ of 500 samples—drawn from the same data distribution but never used during the original training of $M_{\text{base}}$.

For each prefix–suffix pair $(p, s)$, where the suffix $s$ consists of the subsequent 64 tokens, both $M_{\text{base}}$ and $M_{\text{ft}}$ are prompted with:

```
--max-new-tokens=64
```

and their responses are recorded. Let $y_{\text{base}} = M_{\text{base}}(p)$ and $y_{\text{ft}} = M_{\text{ft}}(p)$. We compute the BERT similarity $\text{BERT}(y_{\text{base}}, y_{\text{ft}})$ for each sample in $D^{\text{test}}$ and $D^{\text{hold}}$. To determine whether fine-tuning induces a statistically significant shift in generation behavior, we conduct a one-sided $t$-test on the distributions of BERT similarities from $D^{\text{test}}$ versus $D^{\text{hold}}$, producing a $p$-value at each epoch. The resulting $p$-values are shown in Figure 10 indicate a very poor performance (only one true positive after 1st or 3rd epoch and up to three after 10th epoch compared to 8 true positives for the simple Baseline).

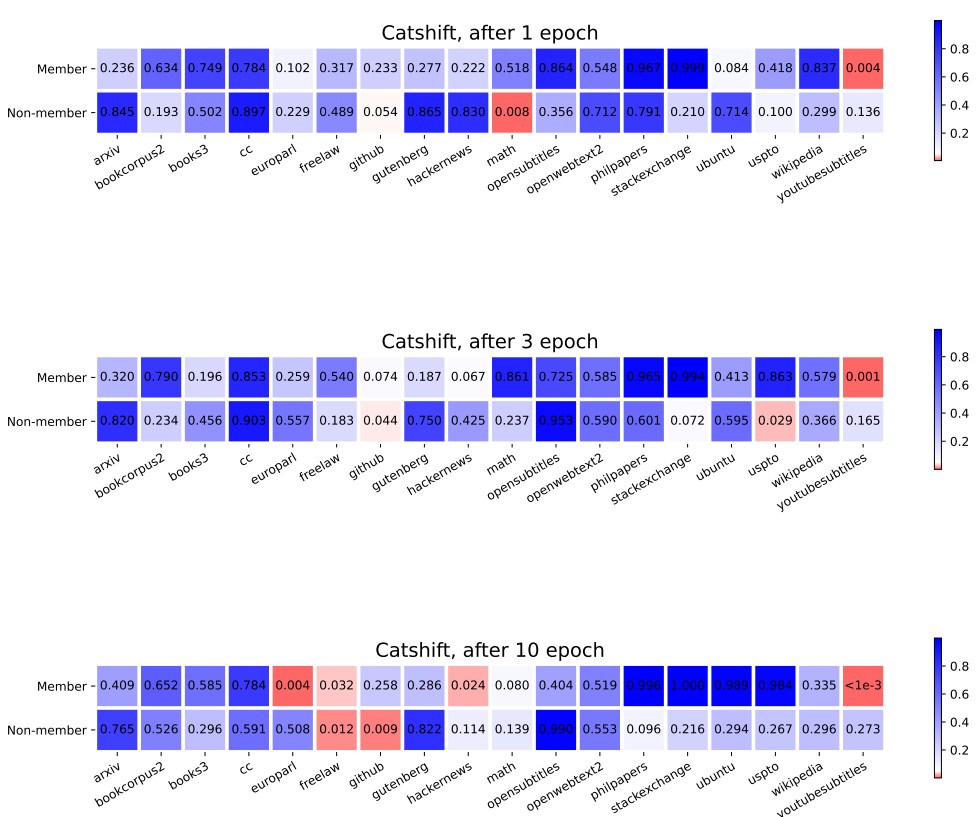

*Figure 10.* **Epoch-wise $p$-values for the CatShift Method.** We show the values after fine-tuning Pythia-410M-deduped (after 1st, 3rd, and 10th epoch). Similar trends are observed across intermediate epochs.

## C. Nonparametric Two-Sample Testing by Betting

An *e-value* is a nonnegative statistic whose expectation under the null hypothesis is at most one. Intuitively, it quantifies evidence against the null: large e-values suggest the null is implausible, while small values are consistent with it. Following the betting framework of Shafer (2021); Shekhar & Ramdas (2023), hypothesis testing can be viewed as a *betting game* against the null. At each round $t$, the arbitrator observes a pair of data points $Z_t = (X_t, Y_t)$ and selects a payoff function $S_t$ with null-conditional expectation at most one. The bettor's wealth is updated multiplicatively as

$$W_t = W_{t-1} S_t(Z_t), \quad W_0 = 1. \tag{10}$$

The resulting sequence $\{W_t\}_{t \geq 0}$ is a nonnegative martingale under $H_0$, known as an *e-process* (Ramdas et al., 2020; Ramdas & Wang, 2025). By Ville's inequality (Ville, 1939), exceeding a threshold (e.g., $W_t \geq 1/\alpha$) guarantees a valid level-$\alpha$ test at any stopping time. Thus, the current wealth $W_t$ itself serves as an e-value, providing a calibrated measure of evidence against the null hypothesis. Such approach introduces numerous advantages over traditional p-value based testing, in particular intuitive properties allowing for simple interpretation of the results.

### C.1. Betting Strategies

Similarly to how there are better and worse estimators, the betting test may be better or worse based on the employed *betting strategy*. A betting strategy is a predictable, data-driven rule that generates stakes from past data, *i.e.,* dictates the fraction of wealth that we bet in each round, based on our previous experience in the game.

We formalize past information via the filtration $(\mathcal{F}_t)_{t \geq 0}$, where for the sequence of data points $\{Z_t\}_{t \geq 1}$ (in our two-sample setting, $Z_t = (X_t, Y_t)$) we set filtration $\mathcal{F}_t = \sigma(Z_1, \ldots, Z_t)$ given by a $\sigma$-algebra on $\{Z_i\}_{i \leq t}$, and take trivial $\mathcal{F}_0$. A rule is *predictable* if it is $\mathcal{F}_{t-1}$-measurable.

Let $\mathcal{Z} = \mathcal{X} \times \mathcal{X}$ denote the observation space. A betting strategy is a sequence of $\mathcal{F}_{t-1}$-measurable maps

$$\mathcal{A}_{\text{bet}} = \{\lambda_t : \mathcal{Z}^{t-1} \to [-\lambda_{\max}, \lambda_{\max}]\}_{t \geq 1}.$$

Given a predictable score function $\tilde{g}_t$, we define the (scalar) round-$t$ score

$$u_t := \tilde{g}_t(Z_t) \in [-1, 1],$$

chosen so that under $H_0$, $\boldsymbol{E}[u_t \mid \mathcal{F}_{t-1}] \leq 0$.

In general, any payoff $S_t$ with $\boldsymbol{E}[S_t \mid \mathcal{F}_{t-1}] \leq 1$ yields valid e-values. A standard and convenient choice is the linearized form

$$S_t(Z_t) = 1 + \lambda_t u_t,$$

which cleanly separates *what* we bet on ($u_t$; produced by the prediction/witness rule) from *how much* we bet ($\lambda_t$; produced by the betting strategy). Linearized form ensures that $S_t \geq 0$ when $|u_t| \leq 1$ and $|\lambda_t| \leq 1$, and simplifies analysis and implementation.

**Log-Optimal Motivation and Stopping Rule.** A good betting strategy is a one that ensures quick growth of evidence under the alternative. This can be quantified by log-optimal principles (Kelly, 1956) and their modern developments (Shafer, 2021; Waudby-Smith & Ramdas, 2024; Grünwald et al., 2024). Motivated by those principles, we focus on strategies that approximately maximize expected log-wealth. Such strategies are conservative in the sense that they never stake all capital and therefore avoid ruin (zero wealth). The resulting wealth process $W_t^\star$ is a (test) martingale under $H_0$ and grows at an exponential rate under fixed alternatives. We adopt the stopping time

$$\tau := \inf\{t \geq 1 : W_t \geq 1/\alpha\},$$

which, by Ville's inequality (Ville, 1939), induces a level-$\alpha$ sequential test.

Exponential wealth growth under fixed alternatives is particularly valuable for dataset inference, where evidence may need to be established from few examples (*e.g.,* in legal settings) or when computing many membership features is costly. When an effect is present, the expected log-wealth grows approximately linearly in $t$, so the wealth $\{W_t\}$ increases at an exponential rate. Consequently, the stopping time $\tau = \inf\{t \geq 1 : W_t \geq 1/\alpha\}$ can be small, meaning fewer observations are required to reach a level-$\alpha$ decision—while anytime validity is preserved by the betting construction and Ville's inequality.

**RKHS and the Witness.** Let $K : \mathcal{X} \times \mathcal{X} \to \mathbb{R}$ be a positive definite kernel. The reproducing kernel Hilbert space (RKHS) $\mathcal{H}_K$ is the completion of finite linear combinations of $\{K(x, \cdot) : x \in \mathcal{X}\}$ with inner product satisfying the reproducing property

$$g(x) = \langle g, K(x, \cdot) \rangle_{\mathcal{H}_K}, \qquad \langle K(x, \cdot), K(y, \cdot) \rangle_{\mathcal{H}_K} = K(x, y).$$

We use the RKHS unit ball $\{g : \|g\|_{\mathcal{H}_K} \le 1\}$ as the witness class underlying kernel MMD. In our sequential procedure, the prediction strategy maintains a predictable witness $g_t \in \mathcal{H}_K$; given the round-$t$ pair $Z_t = (X_t, Y_t)$, the per-round edge is

$$v_t = g_t(X_t) - g_t(Y_t) = \langle g_t, K(X_t, \cdot) - K(Y_t, \cdot) \rangle_{\mathcal{H}_K}.$$

**Kernel MMD.** Kernel maximum mean discrepancy (MMD) underlies a widely used batch two-sample test (Gretton et al., 2012). Sequential nonparametric counterparts exist (Balsubramani & Ramdas, 2016; Manole & Ramdas, 2023), and a concrete sequential kernel-MMD construction with strong guarantees is developed by Shekhar & Ramdas (2023, Sec. 4), which we adopt here as part of our prediction strategy.

We view two-sample distance through the lens of an integral probability metric. For distributions $P_X$ and $P_Y$ on $\mathcal{X}$ with $X \sim P_X$ and $Y \sim P_Y$,

$$d_{\mathcal{G}}(P_X, P_Y) = \sup_{g \in \mathcal{G}} \left( \boldsymbol{E}[g(X)] - \boldsymbol{E}[g(Y)] \right).$$

Let $K : \mathcal{X} \times \mathcal{X} \to \mathbb{R}$ be a positive definite kernel with reproducing kernel Hilbert space (RKHS) $\mathcal{H}_K$. Specializing $\mathcal{G}$ to the RKHS unit ball yields the *kernel MMD*:

$$\mathrm{MMD}(P_X, P_Y) = \sup_{\|g\|_{\mathcal{H}_K} \le 1} \left( \boldsymbol{E}[g(X)] - \boldsymbol{E}[g(Y)] \right) = \|\mu_{P_X} - \mu_{P_Y}\|_{\mathcal{H}_K},$$

where $\mu_P := \boldsymbol{E}[K(X, \cdot)] \in \mathcal{H}_K$ is the kernel mean embedding.

We update the witness online via projected averaging (a special case of projected gradient ascent):

$$g_{t+1} \leftarrow \Pi_{\|g\|_{\mathcal{H}_K} \le 1} \left( g_t + \tfrac{1}{t} \left( K(X_t, \cdot) - K(Y_t, \cdot) \right) \right), \qquad t \ge 1,$$

where $\Pi$ denotes projection onto the RKHS unit ball. The scalar $v_t$ (or a bounded version $u_t \in [-1, 1]$) then feeds the linearized payoff $S_t(Z_t) = 1 + \lambda_t u_t$ used in the wealth update.

**Kolmogorov–Smirnov Discrepancy.** We consider a transformation $T : \mathcal{Z} \to \mathcal{Z}$ such that the null distribution remains invariant under this transformation, whereas distributions under the alternative are altered by it. Formally, let $\mathcal{P}_{\text{null}}$ and $\mathcal{P}_{\text{alt}}$ denote disjoint classes of probability distributions on the observation space $\mathcal{Z}$. We assume

$$P = P \circ T^{-1} \quad \forall P \in \mathcal{P}_{\text{null}}, \qquad P \neq P \circ T^{-1} \quad \forall P \in \mathcal{P}_{\text{alt}}.$$

This invariance property allows us to frame the hypothesis testing problem in terms of a *discrepancy measure* between $P$ and its transformed version $P \circ T^{-1}$.

To quantify this discrepancy, we employ an Integral Probability Metric (IPM). Given a function class $G = \{g : \mathcal{Z} \to [-\frac{1}{2}, \frac{1}{2}]\}$ is defined as

$$d_G(P, P \circ T^{-1}) := \sup_{g \in G} \left| \mathbb{E}_P[g(Z)] - \mathbb{E}_P[g(TZ)] \right|. \tag{11}$$

The Kolmogorov–Smirnov (KS) discrepancy arises as a special case of equation 11 when $G$ is chosen to be the class of indicator functions

$$G_{\text{KS}} = \left\{ g_u(x) = \mathbf{1}\{x \le u\} : u \in \mathbb{R} \right\}. \tag{12}$$

This yields the classical KS distance

$$d_{G_{\text{KS}}}^-(P_X, P_Y) := \sup_{u \in \mathbb{R}} \left( F_X(u) - F_Y(u) \right). \tag{13}$$

where $F_X$ and $F_Y$ denote the cumulative distribution functions of $P_X$ and $P_Y$, respectively. In our hypothesis test of interest, we compare the distribution of the *suspect set* with that of a *held-out set*. We hypothesize that, under the alternative, the

suspect set exhibits *smaller membership feature values* than the held-out set. This corresponds to a stochastic ordering in which the empirical CDF of the suspect set dominates that of the held-out set. Accordingly, we employ a one-sided Kolmogorov–Smirnov discrepancy and bet on maximizing the difference stated in equation 12.

**Online Newton Step (Staking).** At round $t$, the stake $\lambda_t$ is chosen predictably from past data (i.e., $\mathcal{F}_{t-1}$-measurable). After observing $Z_t$, we compute the bounded score $u_t \in [-1, 1]$ (from the kernel-MMD witness), update a one-dimensional curvature accumulator, and then set the next stake for round $t+1$:

$$z_t = \frac{u_t}{1 + \lambda_t u_t}, \qquad a_t = a_{t-1} + z_t^2, \qquad \lambda_{t+1} = \Pi_{[-\lambda_{\max}, \lambda_{\max}]}\left(\lambda_t + \frac{2}{2 - \log 3} \frac{z_t}{a_t}\right),$$

with initialization $\lambda_1 = 0$ and $a_0 = 1$. Here $\Pi_{[-\lambda_{\max}, \lambda_{\max}]}$ denotes projection onto the stake set.

**Kelly Betting**. Another approach to placing bets is the strategy introduced by (Kelly, 1956), which aims to maximize expected log wealth by allocating bets in proportion to the strength of evidence for each outcome. This principled allocation leads to the highest achievable long-term exponential growth of wealth. The update rule given in equation 14 is quadratic approximation of the Kelly criterion.

$$\lambda_t = \text{clip}\left(\frac{\sum_{j=1}^t u_j}{\sum_{j=1}^t u_j^2 + \varepsilon}, -\lambda_{\max}, \lambda_{\max}\right), \qquad t = 1, \ldots, N-1. \tag{14}$$

### C.2. Anytime Validity and Post-Hoc Significance.

Let $\mathcal{P}_0 = \{(P_X, P_Y) : P_X = P_Y\}$ denote the composite null. For any (possibly data-dependent) stopping time $\tau$ adapted to the data filtration and for any predictable strategies used to form the wealth process $W_t$, the e-process property and Ville's inequality imply

$$\sup_{(P_X, P_Y) \in \mathcal{P}_0} \mathbb{P}_{P_X \times P_Y}\left(\sup_{t \geq 1} W_t \geq 1/\alpha\right) \leq \alpha.$$

Equivalently, the test that rejects at $\tau = \inf\{t \geq 1 : W_t \geq 1/\alpha\}$ satisfies

$$\sup_{(P_X, P_Y) \in \mathcal{P}_0} \mathbb{P}_{P_X \times P_Y}(\tau < \infty) \leq \alpha,$$

so the probability of a false rejection is at most $\alpha$ *no matter when we choose to stop*. Formally, this uniform type-I control is established in Shekhar & Ramdas (2023, Thm. 1).

A key advantages of our framework are *anytime validity* and *post-hoc significance*. As discussed, the arbitrator may monitor the test sequentially and reject the null as soon as $W_t \geq 1/\alpha$ *without inflating type-I error*. By contrast, a classical fixed-horizon $p$-value is valid only for a pre-specified sample size (or stopping rule); repeatedly "peeking" at the data and continuing until $p \leq \alpha$ can inflate type-I error unless one uses explicit sequential corrections (e.g., $\alpha$-spending/group-sequential methods). Moreover, value of $W_t$ can be interpreted regardless of the predefined threshold, meaning that if the evidence extensively exceeded the threshold the arbitrator can claim more significant result or if the evidence did not reach initially assumed level an arbitrator can still decide to reject the null but at lower significance.

In our setting, we therefore use e-values: they provide anytime-valid, post-hoc interpretable evidence and, under fixed alternatives, exhibit approximately linear growth of expected log-wealth (hence exponential growth of $W_t$), yielding strong sample efficiency.

We use a *linearized payoff* built from the per-round kernel-MMD witness score. Concretely, with a predictable witness $g_t \in \mathcal{H}_K$ we set $u_t = g_t(X_t) - g_t(Y_t)$ and $S_t(Z_t) = 1 + \lambda_t u_t$. The anytime-valid type-I control and the associated consistency guarantees for this construction follow from the general theory and its kernel-MMD specialization Shekhar & Ramdas, 2023, Thm. 1; Prop. 3, Sec. 4. The betting strategy chooses the stake $\lambda_t$ via the online Newton step (ONS), predictably with respect to the data filtration $(\mathcal{F}_t)$ and using only past observations.

### C.3. Specialization to Dataset Inference: Our Contributions

We instantiate the nonparametric two-sample betting framework from Appendix C for our DI. In all experiments we use the degree-2 polynomial kernel

$$K(x, y) = (\gamma x^\top y + c)^2, \qquad \gamma = \tfrac{1}{p}, \ c = 1,$$

with $p$ feature dimension. This choice preserves sensitivity to *mean differences* via the linear term (enabled by $c = 1$), the primary signal of interest in DI, while still allowing second-order interactions. We pair this kernel-MMD witness with an ONS staking rule to set the stakes $\lambda_t$ predictably from past data.

**Inherited Guarantees and Implications.** Because our construction matches the conditions in Appendix C, all theoretical guarantees carry over: the resulting test is *anytime-valid* (we may stop at the first time $t$ with $W_t \geq 1/\alpha$ without inflating type-I error), and *post-hoc significant* (we may interpret the results regardless of the threshold), and under fixed alternatives the expected log-wealth grows approximately linearly in $t$ (hence $W_t$ grows exponentially). Practically, this yields strong sample efficiency when data are scarce or computing membership features is costly. Moreover, under $H_0$ the wealth remains near its initial value (the payoff has null-conditional expectation at most one), leading to low false positive rates which is an important property in legal or high-stakes DI settings.

Overall, this specialization, degree-2 kernel for mean sensitivity plus ONS staking, adapts the general sequential MMD test to DI while retaining anytime-valid type-I control and fast evidence accumulation when member/non-member distributions differ.

### C.4. Comparison of BADI against baseline methods

We report Area Under the Curve (AUC) metric for our method and two baseline methods (PETAL, Baseline) across three Pythia models: Pythia-410M, Pythia-6.9B, and Pythia-12B (Table 2). BADI consistently outperforms baseline methods for the majority of Pile subsets. Scaling up target model improves detection. For Pythia-12B AUC is significantly above random guess (50%) for every subset. As show in Figure 11, BADI demonstrates superior performance.

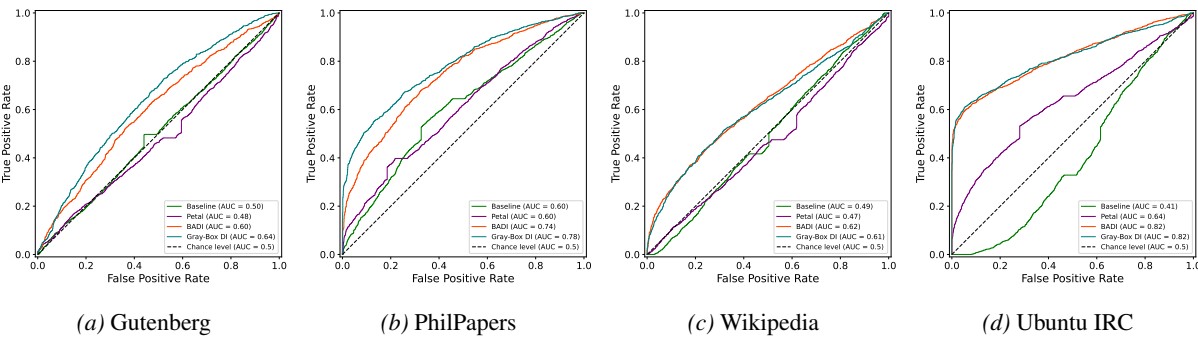

| *(a)* Gutenberg | *(b)* PhilPapers | *(c)* Wikipedia | *(d)* Ubuntu IRC |

*Figure 11.* **ROC curves.** We present comparison of ROC curves for BADI, PETAL, Baseline and Gray-Box DI for the Pythia-12B model.

# D. More Results for Testing by Betting

### D.1. More Results for the Datasets from the Pile

We present extended results to Figure 2 in the following Figure 12 for more datasets from the Pile.

### D.2. More Results across Methods and Model Sizes

We present an extension to Figure 3 on the PhilPpaers dataset in Figure 13.

### D.3. Anytime validity

We increased the sequential horizon from 1000 data points to 2000 to test mean differences in membership features (members vs. non-members). This extension is valid by anytime-validity: the e-process allows continued monitoring without inflating type-I error, and we reject at the first $t$ with $W_t \geq 1/\alpha$ (here $\alpha = 0.05$). Several PILE subsets with low TPR at 1000 data points improve markedly when more data points are available, for example, `hackernews` increases from 0.20 to 0.99. The longer horizon also uncensors runs that did not cross by 1000, allowing us to report additional stopping times; consequently, the mean reported stop time increases, reflecting the inclusion of late (but valid) rejections.

As shown in Figure 14, we plot mean ($\pm$95% CI (Confidence Intervals)) wealth trajectories for the `hackernews` subset.

*Table 2.* **AUC results for dataset inference across model sizes.** Dataset inference performance on the Pile dataset for Baseline, PETAL, Gray-Box DI, and BADI methods across Pythia-12B, Pythia-6.9B, and Pythia-410M models. In the table, **bold** denotes the highest AUC among black-box methods (Baseline, PETAL, and BADI), while underlining indicates the larger value between BADI and the gray-box DI baseline.

| Dataset | Model | | | | | | | | | | | |
| --- | --- | --- | --- | --- | --- | --- | --- | --- | --- | --- | --- | --- |
| | Pythia-12B | | | | Pythia-6.9B | | | | Pythia-410M | | | |
| | Baseline | PETAL | Gray-Box DI | BADI | Baseline | PETAL | Gray-Box DI | BADI | Baseline | PETAL | Gray-Box DI | BADI |
| ArXiv | 0.515 | 0.526 | 0.535 | **0.538** | 0.513 | **0.524** | 0.529 | 0.512 | **0.518** | 0.516 | 0.524 | 0.511 |
| BookCorpus2 | 0.513 | 0.534 | 0.607 | **0.581** | 0.505 | 0.533 | 0.578 | **0.587** | 0.505 | 0.513 | 0.619 | **0.569** |
| Books3 | 0.511 | 0.555 | 0.615 | **0.609** | 0.509 | 0.550 | 0.616 | **0.596** | 0.490 | 0.530 | 0.584 | **0.555** |
| Pile-CC | 0.516 | 0.526 | 0.575 | **0.541** | 0.505 | **0.523** | 0.565 | 0.510 | 0.516 | **0.517** | 0.532 | 0.502 |
| EuroParl | 0.470 | 0.447 | 0.589 | **0.621** | 0.468 | 0.442 | 0.601 | **0.604** | 0.448 | 0.430 | 0.607 | **0.647** |
| FreeLaw | 0.498 | 0.487 | 0.532 | **0.532** | 0.512 | 0.487 | 0.512 | **0.548** | 0.494 | 0.477 | 0.526 | **0.531** |
| Github | 0.530 | 0.520 | 0.517 | **0.539** | 0.533 | 0.519 | 0.528 | **0.549** | 0.535 | 0.513 | 0.544 | **0.560** |
| Gutenberg | 0.502 | 0.477 | 0.636 | **0.600** | 0.494 | 0.472 | 0.639 | **0.605** | 0.488 | 0.453 | 0.605 | **0.559** |
| HackerNews | 0.472 | 0.458 | 0.568 | **0.572** | 0.475 | 0.456 | 0.563 | **0.579** | 0.494 | 0.451 | 0.571 | **0.582** |
| DM Mathematics | 0.504 | 0.527 | 0.548 | **0.532** | 0.508 | 0.525 | 0.534 | **0.561** | 0.514 | 0.521 | 0.521 | **0.539** |
| OpenSubtitles | **0.512** | 0.499 | 0.541 | 0.497 | 0.497 | 0.492 | 0.525 | **0.523** | **0.502** | 0.492 | 0.501 | 0.489 |
| OpenWebText2 | 0.509 | 0.500 | 0.559 | **0.536** | 0.495 | 0.498 | 0.545 | **0.544** | 0.512 | 0.500 | 0.570 | **0.573** |
| PhilPapers | 0.602 | 0.601 | 0.781 | **0.747** | 0.585 | 0.598 | 0.757 | **0.745** | 0.575 | 0.594 | 0.737 | **0.718** |
| StackExchange | 0.612 | 0.665 | 0.701 | **0.671** | 0.618 | 0.663 | 0.701 | **0.690** | 0.614 | 0.654 | 0.703 | **0.689** |
| Ubuntu IRC | 0.406 | 0.635 | 0.817 | **0.825** | 0.407 | 0.632 | 0.799 | **0.805** | 0.414 | 0.625 | 0.810 | **0.801** |
| USPTO | 0.523 | 0.541 | 0.597 | **0.586** | 0.510 | 0.539 | 0.591 | **0.596** | 0.513 | 0.535 | 0.606 | **0.612** |
| Wikipedia | 0.491 | 0.472 | 0.614 | **0.617** | 0.494 | 0.471 | 0.609 | **0.636** | 0.500 | 0.475 | 0.644 | **0.641** |
| YoutubeSubtitles | 0.419 | 0.349 | 0.668 | **0.688** | 0.416 | 0.348 | 0.674 | **0.687** | 0.412 | 0.344 | 0.702 | **0.701** |

With 1000 data points, the mean wealth remains below the $1/\alpha$ threshold, so only a minority of trials reject the null hypothesis. Increasing to 2000 data points yields substantially more threshold crossings; among rejecting runs, the average stopping time is about 948 observations. By anytime validity, extending the horizon in this way does not inflate type-I error.

A note on false positives in Figure 14. To estimate the false positive rate, we compare non-member vs. non-member streams. Due to limited non-member data, these runs were truncated at 1000 points. This truncation does not undermine our FPR conclusions: under the null, the wealth process is a nonnegative test martingale with $\mathbb{P}(\sup_t W_t \geq 1/\alpha) \leq \alpha$, so wealth does not systematically exceed its initial value and type-I error remains controlled. Empirically, we observe wealth staying near one and rare threshold crossings across datasets, consistent with this guarantee.

### D.4. Additional Information on Generalization to Different Model Sizes

We report performance metrics for our dataset inference framework across three Pythia models: Pythia-410M, Pythia-6.9B, and Pythia-12B (Table 3). The FPR matches the theoretical guarantee, remaining at 1% across datasets and model sizes. Scaling up improves detection on Books3, BookCorpus2, and Gutenberg, reflected by higher TPR and shorter average stopping times. Detection is consistently strong on Ubuntu IRC, StackExchange, PhilPapers, and YouTube Subtitles, with high TPR and early stopping. In contrast, ArXiv, Pile-CC, OpenSubtitles, FreeLaw, and GitHub rarely reach significance within the available 2000 examples. Scale effects are not uniform: for Wikipedia (and in some cases EuroParl and USPTO), larger models yield a weaker signal (lower TPR) despite similar FPR.

### D.5. One-sided testing

In dataset inference, one natural hypothesis is that, under the alternative, the suspect set exhibits *smaller membership feature values* than the held-out reference set. To evaluate this hypothesis, we employ the Kolmogorov–Smirnov (KS) discrepancy as our distance metric and place one-sided bets on the event that the empirical CDF of the suspect set dominates that of the held-out set, as described in Section C.1. The results of the one-sided hypothesis testing is consistent with the other test results presented in the study. We also observe an increase in the TPR of ArXiv, BookCorpus2 and Books3 subsets of The Pile dataset.

The test is implemented using the KS discrepancy together with the ONS betting strategy. Consistent with the other experiments in this study, we run the test with 50 random seeds and report the average TPR, FPR, and FNR at a significance level of $\alpha = 0.05$. The results for the Pythia-12B model are summarized in Table 4.

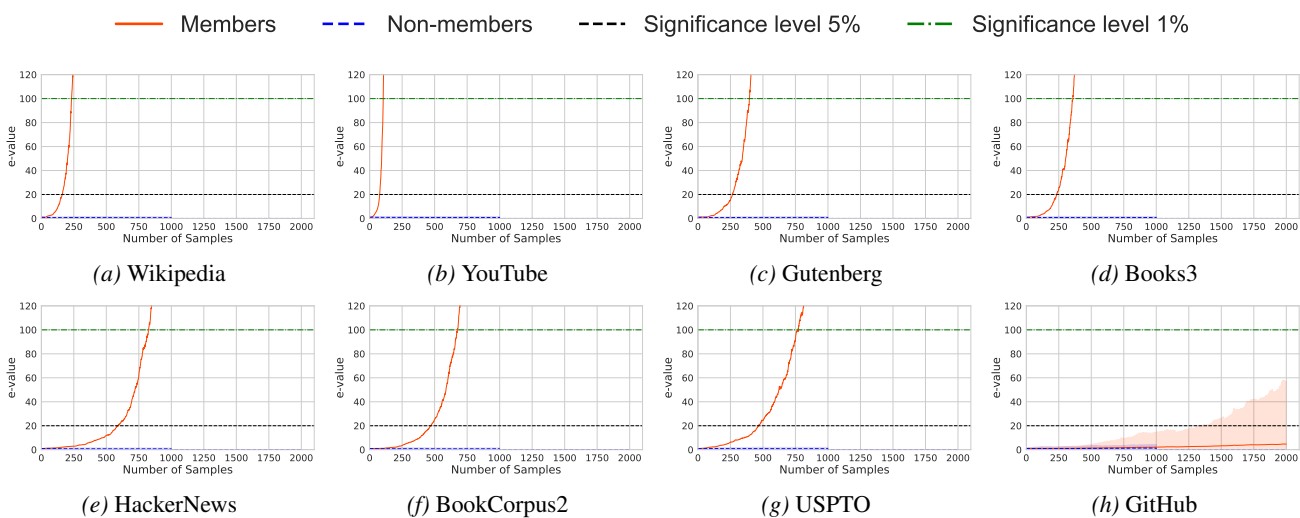

*Figure 12.* **Testing by betting across Pile subsets.** Accumulated wealth trajectories with 95% confidence intervals (orange shading) for sequential testing by betting on the Pythia-12B model (extension of Figure 2).

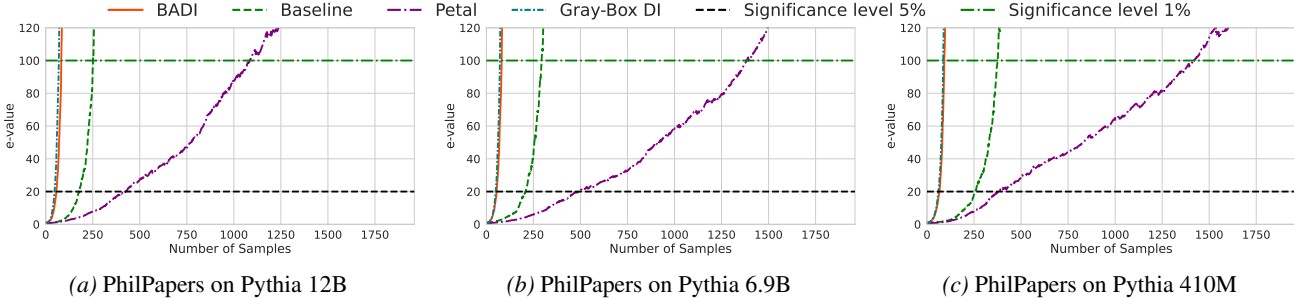

*Figure 13.* **Comparison across methods and model sizes.** We compare BADI against PETAL and Baseline, reporting accumulated wealth for Pythia-12B (left), Pythia-6.9B (middle), and Pythia-410M (right) on and PhilPapers datasets. This is an extension of Figure 3 from the main paper.

### D.6. Ablation on betting strategies

We further evaluate the impact of different betting strategies on the performance of our sequential testing framework. In particular, we consider an approximate Kelly betting strategy, as described in Section C.1. This strategy preserves the same theoretical guarantees as the ONS betting approach, and we empirically observe an exponential growth in wealth under this setup as well.

The experimental configuration mirrors the other experiments presented in this paper. To ensure consistency, we use Kernel-MMD as the pay-off function and vary only the betting strategy. The results of sequential testing with the Kelly strategy for the Pythia-12B model are summarized in Table 5. Our findings indicate that, as long as the theoretical guarantees of the betting framework are preserved, we consistently observe exponential wealth growth, matching the results obtained with the ONS strategy.

### D.7. Ablation on test power

In this subsection, we examine how the choice of betting strategy and associated hyperparameters influences the statistical power of hypothesis testing by betting. In particular, we show that the testing power is largely robust to both the selection of the betting strategy and the choice of the maximum betting cap. Finally, we compare the sequential testing power of BADI with that of classical hypothesis tests based on fixed-sample p-values.

To quantify this robustness, we compute the *sequential power curve* across repeated Monte Carlo simulations. The sequential power at time $t$ is defined as the probability that the test has rejected the null hypothesis on or before time $t$. Formally, for a given $t$, let $T_i$ denote the stopping time for the $i$-th trial. The empirical sequential power is then estimated as:

*Table 3.* **Sequential dataset inference performance across model sizes.** True positive rate (TPR), false positive rate (FPR), and average stopping time for testing by betting across Pile subsets. Results are shown for Pythia-12B (left), Pythia-6.9B (middle), and Pythia-410M (right) using 2000 member/non-member samples and a 5% significance level. N/A indicates that the test did not reach significance within the budget.

| | Model | | | | | | | | |
|---|---|---|---|---|---|---|---|---|---|
| **Dataset** | **Pythia-12B** | | | **Pythia-6.9B** | | | **Pythia-410M** | | |
| | TPR | FPR | Avg. stop | TPR | FPR | Avg. stop | TPR | FPR | Avg. stop |
| ArXiv | $0.05 \pm 0.08$ | $0.00 \pm 0.00$ | N/A | $0.01 \pm 0.02$ | $0.03 \pm 0.08$ | N/A | $0.00 \pm 0.00$ | $0.00 \pm 0.00$ | N/A |
| BookCorpus2 | $1.00 \pm 0.00$ | $0.00 \pm 0.00$ | $521.6 \pm 84.4$ | $0.99 \pm 0.01$ | $0.00 \pm 0.01$ | $634.2 \pm 120.1$ | $0.99 \pm 0.01$ | $0.00 \pm 0.00$ | $578.1 \pm 132.1$ |
| Books3 | $1.00 \pm 0.00$ | $0.01 \pm 0.01$ | $311.8 \pm 58.7$ | $1.00 \pm 0.00$ | $0.00 \pm 0.00$ | $309.0 \pm 57.3$ | $0.97 \pm 0.04$ | $0.00 \pm 0.00$ | $769.2 \pm 171.7$ |
| Pile-CC | $0.08 \pm 0.09$ | $0.00 \pm 0.00$ | N/A | $0.00 \pm 0.00$ | $0.00 \pm 0.00$ | N/A | $0.00 \pm 0.00$ | $0.00 \pm 0.00$ | N/A |
| EuroParl | $0.97 \pm 0.10$ | $0.00 \pm 0.00$ | $345.6 \pm 165.8$ | $0.98 \pm 0.05$ | $0.00 \pm 0.00$ | $330.0 \pm 87.5$ | $1.00 \pm 0.00$ | $0.00 \pm 0.01$ | $233.8 \pm 25.5$ |
| FreeLaw | $0.36 \pm 0.29$ | $0.00 \pm 0.00$ | N/A | $0.48 \pm 0.28$ | $0.00 \pm 0.00$ | N/A | $0.13 \pm 0.16$ | $0.00 \pm 0.00$ | N/A |
| Github | $0.22 \pm 0.29$ | $0.08 \pm 0.15$ | N/A | $0.22 \pm 0.27$ | $0.05 \pm 0.15$ | N/A | $0.33 \pm 0.26$ | $0.00 \pm 0.00$ | N/A |
| Gutenberg | $1.00 \pm 0.00$ | $0.00 \pm 0.01$ | $346.7 \pm 32.0$ | $1.00 \pm 0.00$ | $0.00 \pm 0.01$ | $418.1 \pm 64.5$ | $0.90 \pm 0.13$ | $0.00 \pm 0.00$ | $1039.1 \pm 217.5$ |
| HackerNews | $0.99 \pm 0.01$ | $0.00 \pm 0.00$ | $597.7 \pm 110.0$ | $0.99 \pm 0.01$ | $0.00 \pm 0.00$ | $562.8 \pm 74.8$ | $0.99 \pm 0.03$ | $0.00 \pm 0.00$ | $539.2 \pm 167.4$ |
| DM Mathematics | $0.06 \pm 0.06$ | $0.00 \pm 0.00$ | N/A | $0.29 \pm 0.30$ | $0.00 \pm 0.00$ | N/A | $0.01 \pm 0.01$ | $0.00 \pm 0.00$ | N/A |
| OpenSubtitles | $0.01 \pm 0.01$ | $0.00 \pm 0.00$ | N/A | $0.01 \pm 0.05$ | $0.03 \pm 0.06$ | N/A | $0.00 \pm 0.00$ | $0.00 \pm 0.00$ | N/A |
| OpenWebText2 | $0.18 \pm 0.22$ | $0.00 \pm 0.00$ | N/A | $0.25 \pm 0.26$ | $0.00 \pm 0.00$ | N/A | $0.86 \pm 0.18$ | $0.00 \pm 0.00$ | $1116.3 \pm 261.7$ |
| PhilPapers | $1.00 \pm 0.00$ | $0.00 \pm 0.00$ | $59.3 \pm 1.8$ | $1.00 \pm 0.00$ | $0.00 \pm 0.00$ | $59.6 \pm 2.9$ | $1.00 \pm 0.00$ | $0.00 \pm 0.01$ | $69.9 \pm 2.8$ |
| StackExchange | $1.00 \pm 0.00$ | $0.00 \pm 0.00$ | $106.3 \pm 7.8$ | $1.00 \pm 0.00$ | $0.00 \pm 0.00$ | $107.3 \pm 6.8$ | $1.00 \pm 0.00$ | $0.00 \pm 0.00$ | $106.9 \pm 7.8$ |
| Ubuntu IRC | $1.00 \pm 0.00$ | $0.00 \pm 0.00$ | $31.2 \pm 0.5$ | $1.00 \pm 0.00$ | $0.02 \pm 0.04$ | $31.1 \pm 0.5$ | $1.00 \pm 0.00$ | $0.00 \pm 0.01$ | $33.2 \pm 0.7$ |
| USPTO | $0.99 \pm 0.01$ | $0.00 \pm 0.01$ | $632.5 \pm 108.8$ | $0.99 \pm 0.01$ | $0.00 \pm 0.00$ | $607.5 \pm 78.2$ | $0.99 \pm 0.03$ | $0.00 \pm 0.00$ | $509.5 \pm 126.2$ |
| Wikipedia | $1.00 \pm 0.00$ | $0.00 \pm 0.00$ | $213.1 \pm 17.2$ | $1.00 \pm 0.00$ | $0.00 \pm 0.00$ | $203.9 \pm 15.9$ | $1.00 \pm 0.00$ | $0.01 \pm 0.01$ | $176.3 \pm 14.5$ |
| YoutubeSubtitles | $1.00 \pm 0.00$ | $0.00 \pm 0.00$ | $82.3 \pm 3.8$ | $1.00 \pm 0.00$ | $0.00 \pm 0.00$ | $85.0 \pm 2.9$ | $1.00 \pm 0.00$ | $0.01 \pm 0.02$ | $77.1 \pm 4.1$ |

$$\text{Power}(t) = \frac{1}{\text{num. trials}} \sum_{i=1}^{\text{num. trials}} \mathbf{1}\{T_i \leq t\}.$$

As shown in Figure 15, we plot the sequential power curves for the Ubuntu subset of the Pile dataset using MI features extracted by BADI on Pythia-12B model. The results demonstrate that the hypothesis test is robust to both the choice of betting strategy and the maximum betting cap. In accordance with log-optimality constraints, the maximum betting value must lie within the interval $[-1, 1]$ to prevent the wealth process from collapsing to zero. Within this feasible range, both the ONS and approximate Kelly strategies achieve their highest empirical power at a maximum betting value of $0.5$. However, the sequential power remains stable across different cap values, consistently converging to the significance threshold under the alternative hypothesis.

We additionally compare the power of the sequential testing procedure described above with that of a classical fixed-sample permutation test. As in the sequential setting, we consider aggregated membership scores for the suspect and held-out sets, denoted by $X$ and $Y$, respectively, and use the same kernel maximum mean discrepancy (MMD) with a polynomial kernel as the measure of distributional distance.

A classical *two-sample permutation test with kernel MMD* assesses whether two collections of samples, $X \sim P$ and $Y \sim Q$, are drawn from the same distribution ($H_0 : P = Q$). The test uses the *maximum mean discrepancy (MMD)* as a discrepancy statistic and approximates its null distribution via *label permutations*. We compute an observed test statistic that increases when $P \neq Q$ using the *biased squared MMD* estimator:

$$\widehat{\text{MMD}}^2(X, Y) = \frac{1}{n^2} \sum_{i,i'} k(x_i, x_{i'}) + \frac{1}{n^2} \sum_{j,j'} k(y_j, y_{j'}) - \frac{2}{n^2} \sum_{i,j} k(x_i, y_j).$$

Under the null hypothesis, the pooled sample $Z = X \cup Y$ is *exchangeable* with respect to the labels indicating membership in $X$ or $Y$. To approximate the null distribution of the test statistic, we repeatedly (with $B = 10$ permutations in our experiments) draw a random permutation of the pooled indices, split the permuted samples into two groups of equal size $n$ to form pseudo-samples $(X^{(b)}, Y^{(b)})$, and recompute the squared MMD statistic $T^{(b)} = \widehat{\text{MMD}}^2(X^{(b)}, Y^{(b)})$. The permutation p-value is then defined as the fraction of permuted statistics at least as large as the observed value,

$$\hat{p} = \frac{1 + \sum_{b=1}^{B} \mathbf{1}\{T^{(b)} \geq T_{\text{obs}}\}}{B + 1},$$

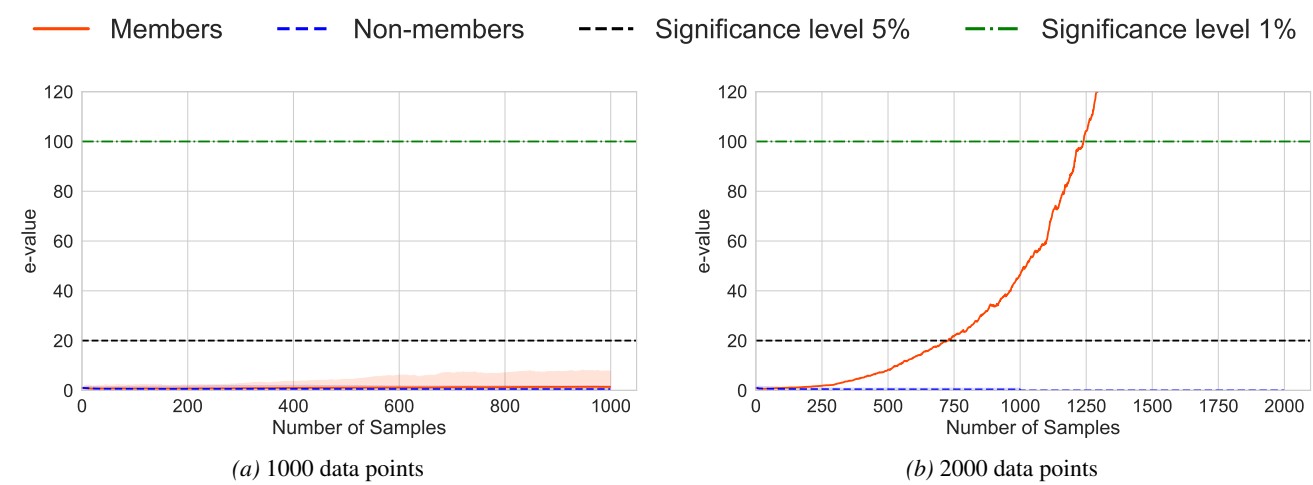

*(a)* 1000 data points  *(b)* 2000 data points

*Figure 14.* **Wealth Trajectories.** We present the wealth trajectories with 95% confidence intervals for `hackernews` (left) with test performed on 1000 data points and (right) test performed on 2000 data points of our dataset inference method and Pythia-410M model. Due to anytime validity of sequential testing, we can add extra data points to the test and eventually cross the desired threshold in the case of true positives.

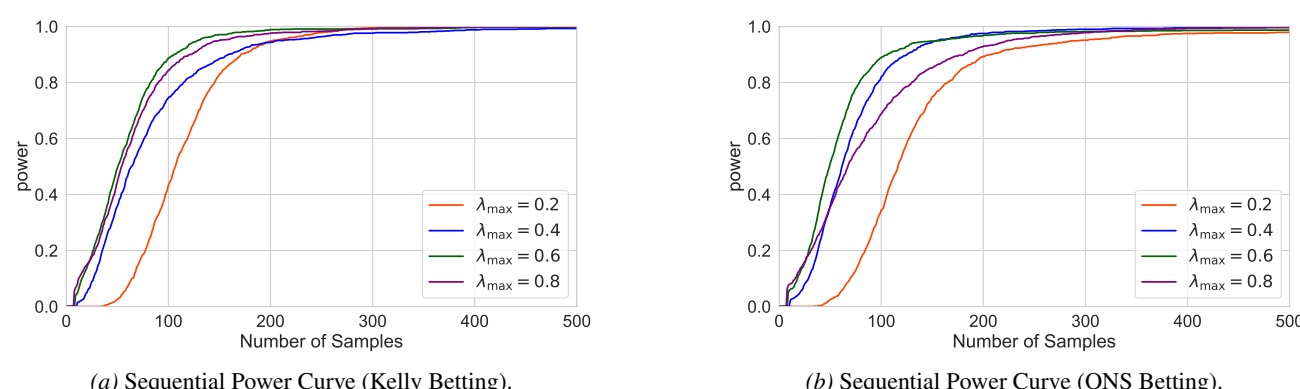

*(a)* Sequential Power Curve (Kelly Betting).  *(b)* Sequential Power Curve (ONS Betting).

*Figure 15.* **Robustness of Sequential Power Across Betting Strategies.** We compare sequential power curves for the approximate Kelly strategy and the ONS strategy on the Ubuntu subset. Results show that detection performance is consistent across strategies.

corresponding to a one-sided test in which larger MMD values provide stronger evidence against $H_0$. The null hypothesis is rejected at significance level $\alpha$ if $\hat{p} \leq \alpha$.

This testing procedure is fully nonparametric and relies solely on exchangeability under the null hypothesis. With a sufficient number of permutations, it provides an accurate approximation to the null distribution of the MMD statistic. To estimate statistical power, we repeat the entire procedure across multiple Monte Carlo trials (we use 20 trials) and over increasing sample sizes. The empirical power at a given sample size $n$ is then defined as the fraction of trials in which the permutation p-value falls below the chosen significance level $\alpha$.

The results of this comparison are shown in Figure 16, which reports empirical power as a function of sample size for both the sequential betting-based test and the classical permutation test. The permutation test is evaluated only at predetermined sample sizes and therefore exhibits a delayed increase in power, reflecting its fixed-sample nature. In contrast, the sequential test accumulates evidence continuously over observations, leading to a smooth and monotonic increase in power as more samples are observed. Across all datasets, the sequential procedure consistently reaches high power, and in particular power close to one, at substantially smaller sample sizes than the permutation test. This demonstrates that sequential testing enables earlier and more sample-efficient rejection of the null hypothesis while maintaining valid Type-I error control under optional stopping, whereas the classical permutation test is only valid when evaluated at a fixed, pre-specified sample size.

*Table 4.* **One-sided vs. two-sided hypothesis testing for Pythia-12B.** Comparison of one-sided and two-sided hypothesis testing setups for sequential dataset inference across Pile subsets. Results are reported using 2000 member/non-member samples at a significance level of $\alpha = 0.05$. N/A indicates that the test did not reach significance within the budget.

| | Hypothesis Testing Setup | | | | | |
| --- | --- | --- | --- | --- | --- | --- |
| | One-sided | | | Two-sided | | |
| Dataset | TPR | FPR | Avg. stop | TPR | FPR | Avg. stop |
| ArXiv | $0.14 \pm 0.35$ | $0.00 \pm 0.00$ | N/A | $0.05 \pm 0.08$ | $0.00 \pm 0.00$ | N/A |
| BookCorpus2 | $1.00 \pm 0.00$ | $0.02 \pm 0.14$ | $167.0 \pm 112.6$ | $1.00 \pm 0.00$ | $0.00 \pm 0.00$ | $521.6 \pm 84.4$ |
| Books3 | $1.00 \pm 0.00$ | $0.04 \pm 0.20$ | $343.6 \pm 269.4$ | $1.00 \pm 0.00$ | $0.01 \pm 0.01$ | $311.8 \pm 58.7$ |
| Pile-CC | $0.02 \pm 0.14$ | $0.02 \pm 0.14$ | N/A | $0.08 \pm 0.09$ | $0.00 \pm 0.00$ | N/A |
| EuroParl | $1.00 \pm 0.00$ | $0.00 \pm 0.00$ | $234.3 \pm 120.8$ | $0.97 \pm 0.10$ | $0.00 \pm 0.00$ | $345.6 \pm 165.8$ |
| FreeLaw | $0.24 \pm 0.43$ | $0.04 \pm 0.20$ | N/A | $0.36 \pm 0.29$ | $0.00 \pm 0.00$ | N/A |
| Github | $0.84 \pm 0.37$ | $0.04 \pm 0.20$ | $1328.0 \pm 584.4$ | $0.22 \pm 0.29$ | $0.08 \pm 0.15$ | N/A |
| Gutenberg | $1.00 \pm 0.00$ | $0.02 \pm 0.14$ | $272.6 \pm 159.5$ | $1.00 \pm 0.00$ | $0.00 \pm 0.01$ | $346.7 \pm 32.0$ |
| HackerNews | $1.00 \pm 0.00$ | $0.04 \pm 0.20$ | $543.7 \pm 315.1$ | $0.99 \pm 0.01$ | $0.00 \pm 0.00$ | $597.7 \pm 110.0$ |
| DM Mathematics | $0.04 \pm 0.20$ | $0.02 \pm 0.14$ | N/A | $0.06 \pm 0.06$ | $0.00 \pm 0.00$ | N/A |
| OpenSubtitles | $0.02 \pm 0.14$ | $0.00 \pm 0.00$ | N/A | $0.01 \pm 0.01$ | $0.00 \pm 0.00$ | N/A |
| OpenWebText2 | $0.18 \pm 0.39$ | $0.02 \pm 0.14$ | N/A | $0.18 \pm 0.22$ | $0.00 \pm 0.00$ | N/A |
| PhilPapers | $1.00 \pm 0.00$ | $0.02 \pm 0.14$ | $43.8 \pm 28.2$ | $1.00 \pm 0.00$ | $0.00 \pm 0.00$ | $59.3 \pm 1.8$ |
| StackExchange | $1.00 \pm 0.00$ | $0.04 \pm 0.20$ | $77.7 \pm 54.7$ | $1.00 \pm 0.00$ | $0.00 \pm 0.00$ | $106.3 \pm 7.8$ |
| Ubuntu IRC | $1.00 \pm 0.00$ | $0.00 \pm 0.00$ | $25.1 \pm 14.8$ | $1.00 \pm 0.00$ | $0.00 \pm 0.00$ | $31.2 \pm 0.5$ |
| USPTO | $1.00 \pm 0.00$ | $0.04 \pm 0.20$ | $475.0 \pm 315.4$ | $0.99 \pm 0.01$ | $0.00 \pm 0.01$ | $632.5 \pm 108.8$ |
| Wikipedia | $1.00 \pm 0.00$ | $0.04 \pm 0.20$ | $242.1 \pm 135.6$ | $1.00 \pm 0.00$ | $0.00 \pm 0.00$ | $213.1 \pm 17.2$ |
| YoutubeSubtitles | $1.00 \pm 0.00$ | $0.02 \pm 0.14$ | $60.8 \pm 40.0$ | $1.00 \pm 0.00$ | $0.00 \pm 0.00$ | $82.3 \pm 3.8$ |

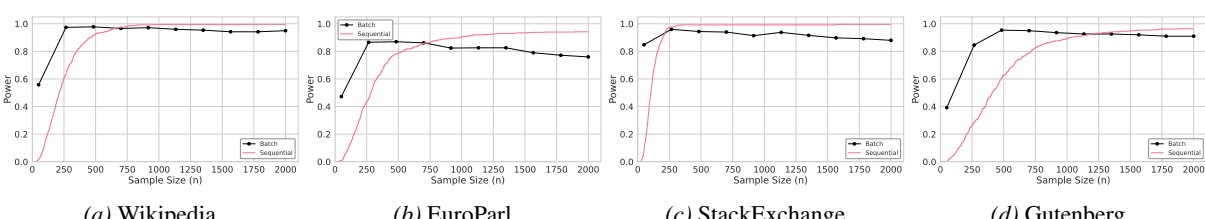

| *(a)* Wikipedia | *(b)* EuroParl | *(c)* StackExchange | *(d)* Gutenberg |

*Figure 16.* **Statistical Power Comparison.** We compare sequential power curves for the sequential hypothesis testing by betting to that of classical two-sample permutation test with kernel-MMD.

## D.8. Robustness to Polluted Held-out Set (Validation Data)

To evaluate the robustness of our sequential testing procedure under held-out-set (validation data) contamination, we conduct an experiment in which a specified proportion of non–members is replaced with true training members. We then apply our hypothesis testing by betting method to the Ubuntu subset of the Pile dataset (Gao et al., 2020) using the Pythia–12B model (Biderman et al., 2023).

In this framework, the bettor adaptively places stakes based on all observations seen so far. Under the alternative hypothesis, accumulated evidence causes the wealth process to grow. We execute the experiment over 50 random seeds and report the average wealth trajectory in Figure 17. For pollution levels up to 70%, the test reliably gathers sufficient evidence to cross the 1% significance threshold within the first 200 observations. At 80% pollution, the rate of wealth growth diminishes, but the trajectory still shows consistent upward movement, indicating that the remaining clean examples provide enough evidence against the null. At 90% pollution, the curves flatten substantially, reflecting that the contaminated distribution becomes nearly indistinguishable from the null.

Overall, the results demonstrate that the betting-based sequential test is robust to substantial contamination of the held-out set (validation data). Even with as much as 70% of the non–members replaced by members, the procedure continues to accumulate evidence and successfully rejects the null hypothesis.

A notable aspect of Figure 17 is that the curve for a 30% pollution ratio reaches the 1% significance threshold with fewer observations than the 20% curve, and even performs comparably to the clean (0%) baseline. At first glance, this appears counter–intuitive, since higher contamination should make the hypothesis test harder. However, this behavior is fully consistent with the properties of the wealth process.

The e–value process is a stochastic, path–dependent quantity. In our experiments, each curve represents an average over 50

*Table 5.* **Ablation on betting strategies for Pythia-12B.** Comparison of approximate Kelly betting and ONS betting strategies for sequential dataset inference across Pile subsets. Results are reported using 2000 member/non-member samples at a significance level of $\alpha = 0.05$. N/A indicates that the test did not reach significance within the budget.

| | Betting Strategy | | | | | |
| Dataset | Kelly Betting | | | ONS Betting | | |
| | TPR | FPR | Avg. stop | TPR | FPR | Avg. stop |
| --- | --- | --- | --- | --- | --- | --- |
| ArXiv | $0.03 \pm 0.02$ | $0.02 \pm 0.01$ | N/A | $0.05 \pm 0.08$ | $0.00 \pm 0.00$ | N/A |
| BookCorpus2 | $0.95 \pm 0.07$ | $0.02 \pm 0.02$ | $528.8 \pm 180.2$ | $1.00 \pm 0.00$ | $0.00 \pm 0.00$ | $521.6 \pm 84.4$ |
| Books3 | $0.81 \pm 0.16$ | $0.02 \pm 0.01$ | $852.1 \pm 224.8$ | $1.00 \pm 0.00$ | $0.01 \pm 0.01$ | $311.8 \pm 58.7$ |
| Pile-CC | $0.03 \pm 0.02$ | $0.02 \pm 0.01$ | N/A | $0.08 \pm 0.09$ | $0.00 \pm 0.00$ | N/A |
| EuroParl | $0.69 \pm 0.18$ | $0.02 \pm 0.02$ | $1001.7 \pm 240.0$ | $0.97 \pm 0.10$ | $0.00 \pm 0.00$ | $345.6 \pm 165.8$ |
| FreeLaw | $0.04 \pm 0.03$ | $0.02 \pm 0.01$ | N/A | $0.36 \pm 0.29$ | $0.00 \pm 0.00$ | N/A |
| GitHub | $0.09 \pm 0.06$ | $0.02 \pm 0.02$ | N/A | $0.22 \pm 0.29$ | $0.08 \pm 0.15$ | N/A |
| Gutenberg | $0.59 \pm 0.19$ | $0.03 \pm 0.03$ | $1257.5 \pm 230.0$ | $1.00 \pm 0.00$ | $0.00 \pm 0.01$ | $346.7 \pm 32.0$ |
| HackerNews | $0.55 \pm 0.23$ | $0.02 \pm 0.01$ | $1293.9 \pm 265.7$ | $0.99 \pm 0.01$ | $0.00 \pm 0.00$ | $597.7 \pm 110.0$ |
| DM Mathematics | $0.04 \pm 0.05$ | $0.02 \pm 0.01$ | N/A | $0.06 \pm 0.06$ | $0.00 \pm 0.00$ | N/A |
| OpenSubtitles | $0.02 \pm 0.02$ | $0.02 \pm 0.02$ | N/A | $0.01 \pm 0.01$ | $0.00 \pm 0.00$ | N/A |
| OpenWebText2 | $0.04 \pm 0.02$ | $0.02 \pm 0.02$ | N/A | $0.18 \pm 0.22$ | $0.00 \pm 0.00$ | N/A |
| PhilPapers | $0.96 \pm 0.07$ | $0.02 \pm 0.02$ | $243.9 \pm 139.1$ | $1.00 \pm 0.00$ | $0.00 \pm 0.00$ | $59.3 \pm 1.8$ |
| StackExchange | $0.99 \pm 0.02$ | $0.02 \pm 0.01$ | $240.6 \pm 77.0$ | $1.00 \pm 0.00$ | $0.00 \pm 0.00$ | $106.3 \pm 7.8$ |
| Ubuntu IRC | $1.00 \pm 0.00$ | $0.02 \pm 0.01$ | $67.0 \pm 13.8$ | $1.00 \pm 0.00$ | $0.00 \pm 0.00$ | $31.2 \pm 0.5$ |
| USPTO | $0.40 \pm 0.21$ | $0.02 \pm 0.01$ | $1530.1 \pm 203.1$ | $0.99 \pm 0.01$ | $0.00 \pm 0.01$ | $632.5 \pm 108.8$ |
| Wikipedia | $0.89 \pm 0.11$ | $0.02 \pm 0.01$ | $621.4 \pm 212.0$ | $1.00 \pm 0.00$ | $0.00 \pm 0.00$ | $213.1 \pm 17.2$ |
| YoutubeSubtitles | $0.97 \pm 0.06$ | $0.02 \pm 0.02$ | $238.5 \pm 118.9$ | $1.00 \pm 0.00$ | $0.00 \pm 0.00$ | $82.3 \pm 3.8$ |

random seeds, where the order of samples is independently shuffled at each run. Because the sequential test operates online, observing data points one at a time, early observations with a large discrepancy between the null and alternative can yield disproportionately large multiplicative jumps in the wealth. Consequently, contamination can occasionally introduce early data points that appear highly informative to the bettor, causing faster boundary crossing even when the overall setting is less favorable.

Importantly, theory does not require that a "stronger" or cleaner alternative always leads to earlier stopping. For e–processes, the only guarantee is that under the alternative the wealth will eventually grow and cross the threshold with high probability; it does not impose monotonicity of stopping times across different alternatives or contamination levels. The variability observed in Figure 17 therefore reflects inherent stochasticity in the sequential evidence accumulation process, rather than a violation of the expected theoretical behavior.

### D.9. Robustness to non-member selection

To further evaluate the robustness of the hypothesis testing by betting framework with respect to the selection of non-member data points, we used the MI scores extracted by our method from the Pythia-12B model on the Ubuntu subset of The Pile dataset. We randomly shuffled the data and divided it into batches of 100 data points each. We then applied the sequential test independently to every batch. The full dataset contains 2,000 member and non-member data points, resulting in 20 subsets of 100 points each. For every subset, we ran the test with 50 random seeds and plotted the average log wealth across seeds.

The results are shown in Figure 18. In every batch, the test successfully accumulates sufficient evidence to reject the null hypothesis and reaches the 1% significance threshold. This demonstrates that the test is robust to the choice of non-member samples: as long as the discrepancy metric provides enough evidence, the betting strategy allocates wagers so that the wealth grows exponentially under the alternative.

## E. Generalization to Different Model Families

**GPT-3.5-Turbo on BookMIA.** We also perform the detection of the BookMIA 2023 dataset (Shi, 2023) on the GPT-3.5-Turbo model. BookMIA serves as a benchmark designed to evaluate MIA methods, specifically in detecting pretraining data from OpenAI models that are released before 2023. As shown in Figure 19, our approach consistently succeeds in this restricted setting: it crosses the 1% significance threshold in fewer than 200 samples and maintains uniform FPR control across 1,000 observations. On average, our method achieves a TPR of 0.995 with an average stopping time of 125.04 observations.

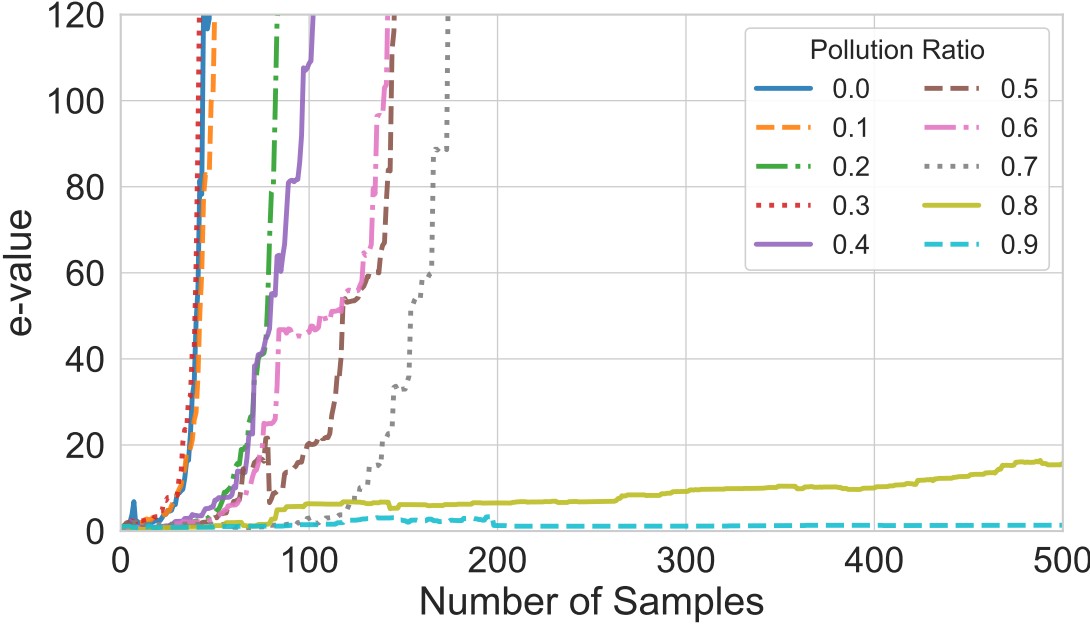

*Figure 17.* **Effect of Pollution Ratio in the Held-Out Set on Sequential e-value Growth.** The plot shows the mean wealth (e-value) trajectories for different non-member pollution ratios ranging from 0 to 0.9. Each curve corresponds to a different injected pollution level. The curves show that once sufficient evidence accumulates against the null hypothesis, the wealth process increases and eventually crosses the significance threshold.

*Table 6.* **AUC results of DI on Dolma dataset and OLMo-7B.** We compare our BADI with PETAL and Baseline methods.

| Dataset | Baseline | PETAL | BADI (Ours) |
|---|---|---|---|
| Gutenberg | 0.664 | 0.586 | **0.801** |
| Common Crawl | 0.531 | **0.619** | 0.612 |
| PeS2o | 0.446 | 0.451 | **0.762** |
| Reddit | 0.482 | 0.432 | **0.555** |
| StackExchange | 0.501 | 0.532 | **0.738** |
| Wikipedia | 0.484 | 0.544 | **0.762** |

**Additional Results for Black-box APIs.** We show the additional ROC curves corresponding to the results in the main paper in Section 5.6 in Figure 20 for the sequential betting-based hypothesis testing.

**OLMo-7B.** We apply the BADI framework to the OLMo-7B model by conducting sequential hypothesis testing via betting. Our evaluation is performed on multiple subsets of the Dolma dataset (Soldaini et al., 2024), and the results are shown in Figure 21. Across diverse data sources (including Reddit, StackExchange, Common Crawl, Wikipedia, Gutenberg, and PeS2o) BADI consistently detects membership, with e-values surpassing 100 for most subsets. Although the Reddit subset does not yet exceed the 5% significance threshold, the e-value exhibits a steadily increasing trend as more data are observed, indicating that the test is on track toward rejection. These findings demonstrate that BADI remains reliable and effective across heterogeneous data distributions and model families. Additionally we compare BADI with PETAL and Baseline method using the AUC metric. The results are reported in Table 6.

**Qwen2-7B.** To further evaluate the robustness of BADI across different model families, we assess our method on both the QWEN2-7B BASE and QWEN2-7B-INSTRUCT (Yang et al., 2024) models using the BOOKMIA 2023 dataset. The results in Figure 22 demonstrate a significant effect in the wealth process, indicating that BOOKMIA data was indeed included in the training set of QWEN2-7B.

Moreover, our method not only successfully infers membership for the BOOKMIA dataset, but also remains effective for instruction-tuned models designed for improved alignment. As shown in Figure 22, the instruction-tuned model requires slightly more observations to reach the 1% significance threshold. We hypothesize that this additional difficulty may be due to perturbations introduced into the model's output distribution during the instruction-tuning process.

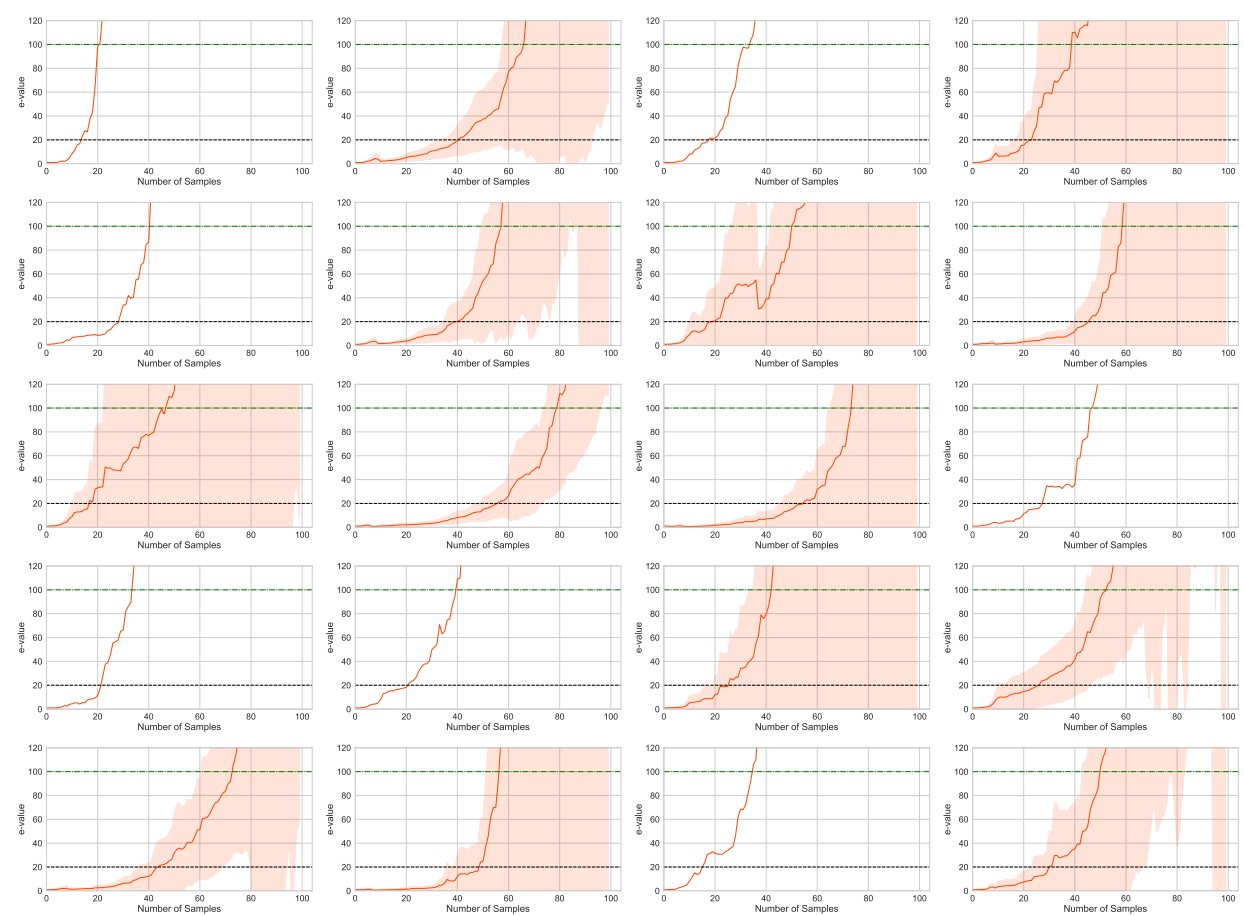

*Figure 18.* **E-process trajectories for 20 batches of 100 data points from the Ubuntu subset of the Pile dataset**. We use MI features collected with our method from the Pythia-12B model. The results show that the betting-based testing framework is robust in distinguishing non-member data points, consistently accumulating sufficient evidence to reject the null hypothesis in each subset independently.

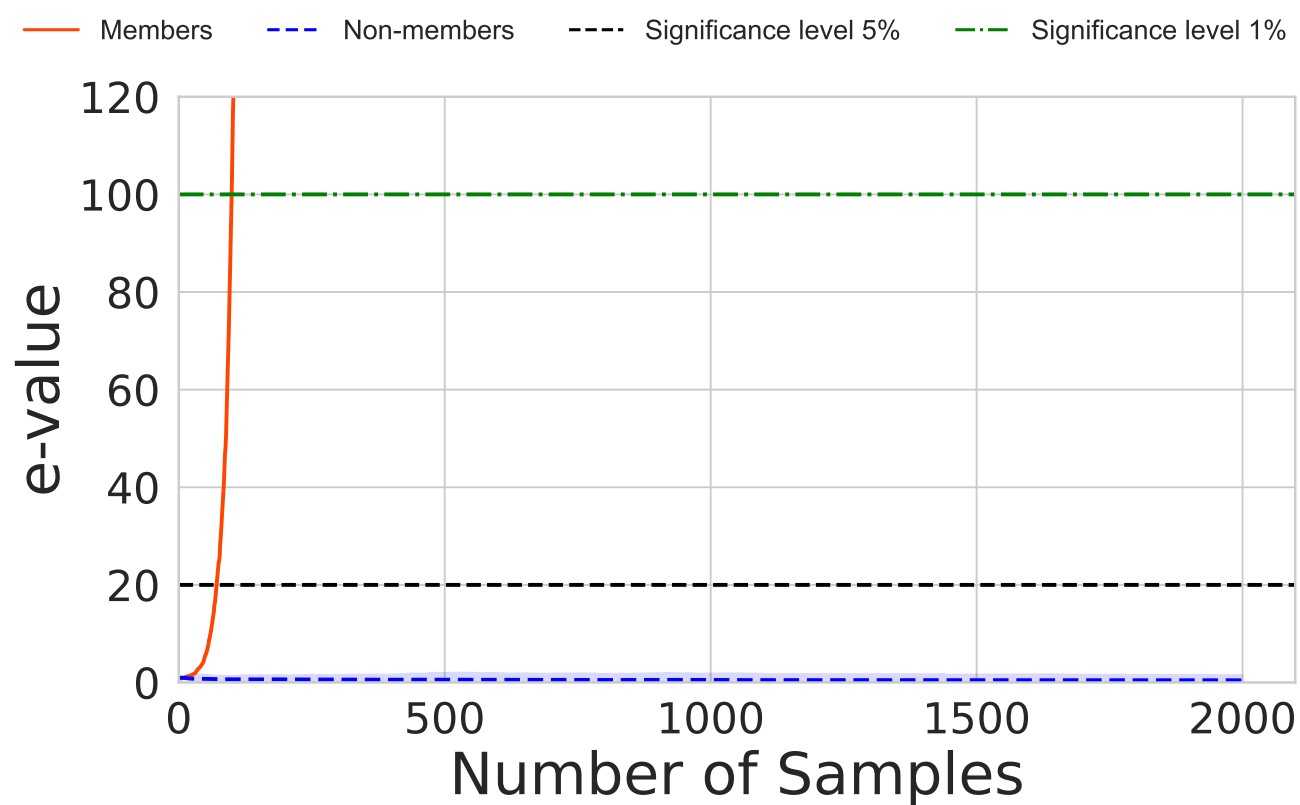

*Figure 19.* **BADI for Black-box LLM APIs.** We present the wealth trajectory for BookMIA dataset with GPT-3.5-Turbo model.

### E.1. Black-Box API Evaluation Details

We apply BADI to black-box large language model (LLM) APIs using the WikiMIA-2024 dataset and detail the evaluation protocol below. WikiMIA-2024 adopts March 2024 as its knowledge cut-off date for distinguishing member and non-member samples. To determine the corresponding knowledge cut-off dates of the evaluated LLMs, we rely on publicly available documentation, primarily the repository at `https://github.com/HaoooWang/llm-knowledge-cutoff-dates`, which collects dates from official technical reports, API providers, GitHub issues, and other public resources.

Notably, the WikiMIA-2024 cut-off date precedes the reported knowledge cut-off for GPT-5. Consequently, a fraction of the validation set may contain samples that are members of GPT-5's training data. However, as shown in Section D.8, BADI is robust to polluted validation sets: even when a subset of validation samples are members, the method remains effective, although additional observations may be required to reach a fixed significance threshold. This effect likely explains the reduced performance observed for GPT-5 in our experiments.

An additional consideration for GPT-5 is its strong instruction tuning. When providing a suffix-only input via OpenAI's Responses API, the model may generate meta-level or instructional outputs rather than a direct text continuation. To mitigate this behavior, we prepend the following prompt to each suffix before querying the model:

> "You are a text-completion engine. Continue the given text seamlessly. Do not add commentary, titles, explanations, quotes, or extra formatting. Output ONLY the continuation."

This prompting strategy is applied exclusively to GPT-5. Other models, including GPT-4, GPT-4o, and GPT-4.1, are queried using the Chat Completions API, while GPT-3.5-Turbo is accessed through the legacy Chat Completions API. Despite these interface differences and the strong instruction-following behavior of newer models, we successfully recover membership signals using BADI by framing queries as text continuation tasks, demonstrating the applicability of BADI in realistic black-box API settings. We a provide an example of a prefix, ground-truth suffix and generations by each black-box LLM in Table 7.

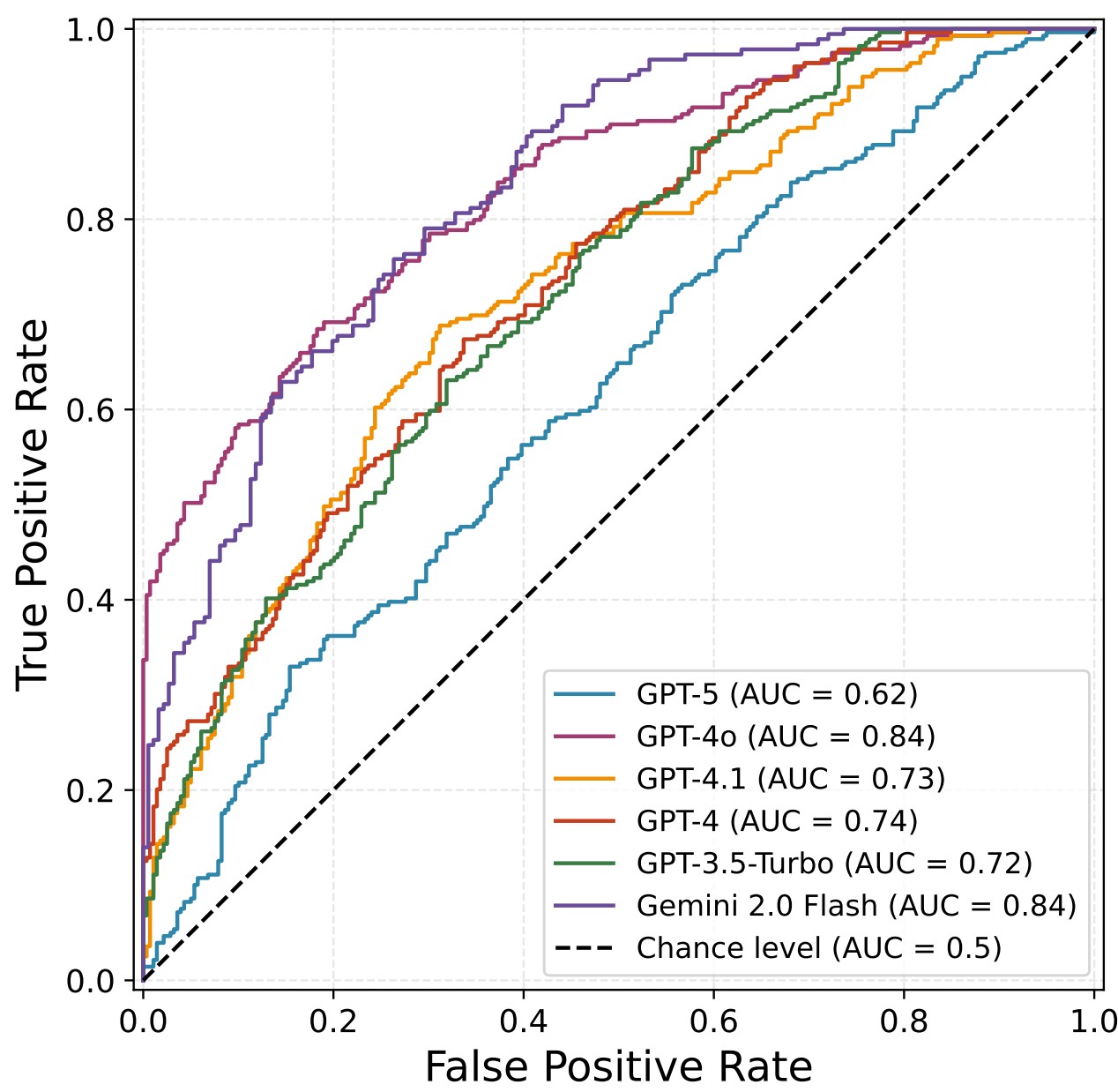

*Figure 20.* **ROC curves of BADI on black-box LLM APIs.** We test the WikiMIA dataset against black-box LLM APIs, demonstrating the effectiveness of BADI for dataset inference in black-box settings.

## F. Sigmoid based Calibration

We investigate the underlying factors contributing to the strong performance of sigmoid-based calibration and seek to explain why, counter-intuitively, this black-box approach outperforms Gray-Box DI on certain subsets of the Pile dataset (Table 2).

To this end, we extract the ground-truth per-token log-probabilities from model generations and compare them against the per-token probability estimates produced by the Baseline, PETAL, and BADI methods. To enable an unconstrained and numerically stable comparison, we map all per-token probabilities to logit space, yielding values in the range $(-\infty, +\infty)$. We then analyze the resulting distributions by plotting density histograms of per-token logit values for member and non-member samples, as shown in Figure 23. The degree of separability between these distributions is quantified using an overlap (OVL) metric.

In addition, to directly assess the accuracy of probability estimation, we compute the mean absolute error (MAE) between

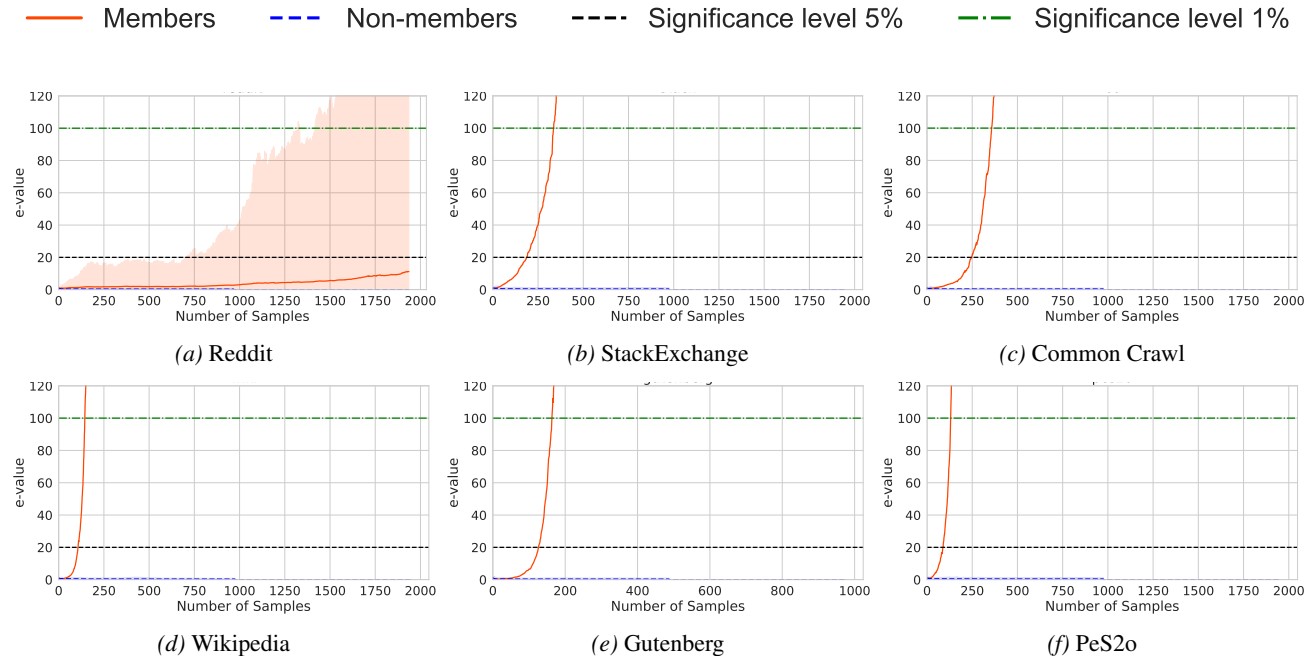

*Figure 21.* **Our Testing by Betting.** We present accumulated wealth with 95% confidence intervals of the e-process performed on various subsets on the OLMo-7B model.

the estimated and ground-truth per-token logits. All experiments are conducted using the Pythia-12B model across 18 subsets of the Pile dataset, with the MAE results summarized in Table 8.

From Figure 23, we observe that PETAL achieves the strongest separation between member and non-member samples, which is consistent with its low mean absolute error (MAE) in per-token probability estimation. However, in the final dataset inference results, sigmoid-based calibration consistently outperforms both PETAL and the Baseline across nearly all Pile subsets (Table 2) and often exhibits evidence accumulation rate similar to that of Gray-Box DI (Figure 13).

This discrepancy can be attributed to how sigmoid-based calibration estimates per-token probabilities. As shown in Figure 23, the true per-token logits follow a heavy-tailed distribution, a behavior that is also captured by PETAL and is further amplified in the Baseline, which produces numerous extreme outlier estimates. These heavy tails adversely affect downstream statistical testing and largely explain the weaker performance of likelihood-based baselines. In contrast, although sigmoid calibration yields higher estimation error and greater overlap between member and non-member distributions, it preserves the correct directional signal while avoiding heavy-tailed behavior. This property explains why it has an on-par performance with gray-box DI.

Furthermore, we adopt sigmoid-based mapping to improve numerical stability during feature normalization and downstream statistical testing. In preliminary experiments, likelihood-based features exhibited a large dynamic range, with negative log-probabilities spanning approximately $[0, 15]$. Under high uncertainty, moderate increases in log-loss (e.g., $\approx 10$) resulted in extremely large perplexity values, which dominated variance estimates during normalization and collapsed other features toward zero. By contrast, the sigmoid transformation yields bounded, well-scaled features, preventing individual samples from disproportionately influencing normalization. This stabilization also mitigates issues arising from reference-model calibration, as used in PETAL, where confidence mismatches between models can amplify spurious signals. Overall, sigmoid-based calibration provides a numerically robust alternative that retains discriminative information while avoiding the pathological effects of heavy-tailed likelihood-based features.

## G. LLM Usage Declaration

We used large language models for refining text, focusing exclusively on style, grammar, and spelling, without compromising the original semantic meaning.

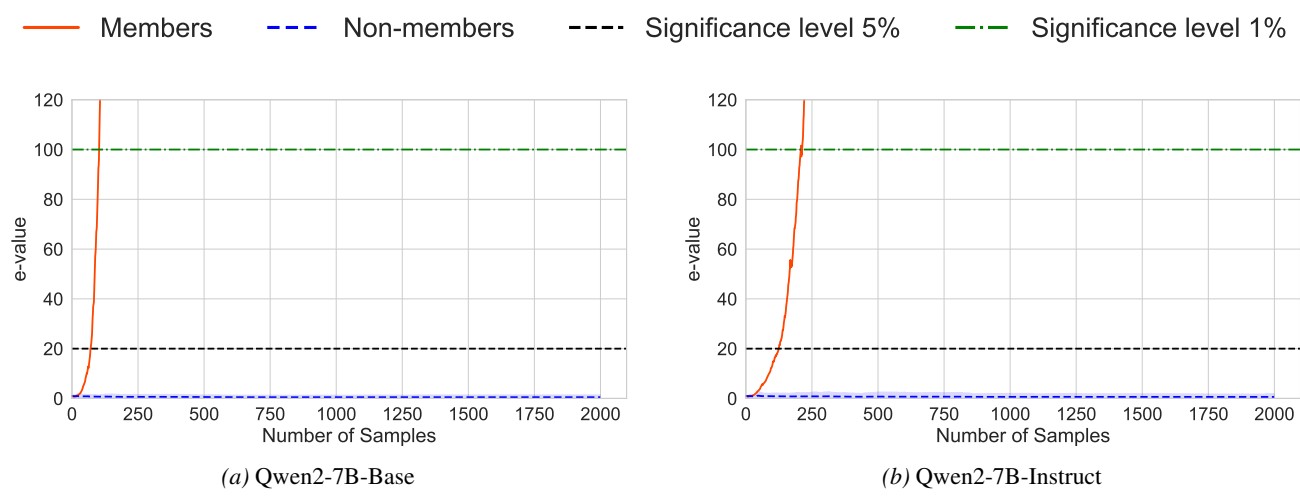

*(a)* Qwen2-7B-Base          *(b)* Qwen2-7B-Instruct

*Figure 22.* **BADI for Qwen2 model variants.** The results show that BADI successfully detects the membership of the BOOKMIA dataset in both models. In addition, the instruction-tuned model requires slightly more observations to reach the $1\%$ significance threshold, suggesting that instruction tuning may introduce perturbations to the output distribution that make membership detection more challenging.

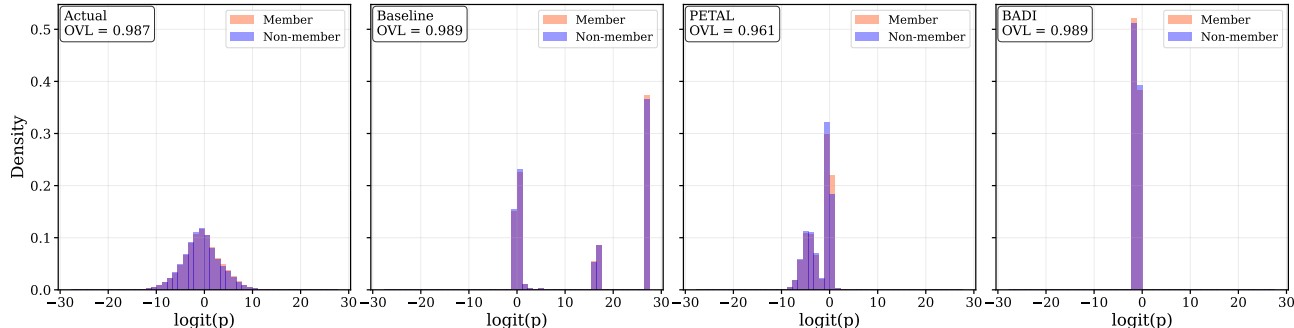

*Figure 23.* **Per-token logit distributions.** This figure shows histograms of per-token probabilities expressed in logit space for member and non-member samples. We quantify the separability of these distributions using the overlap (OVL) metric, which measures the degree of overlap between the two histograms. The OVL value ranges from 0 to 1, where 1 indicates complete overlap and lower values indicate greater distributional separation.

*Table 7.* **Example generations for a fixed prefix.** We show the ground-truth suffix and the continuations generated by each black-box LLM when prompted with the same prefix (text-completion setting).

| Field / Model | Text |
|---|---|
| **Prefix** | Christina Victoria Grimmie (March 12, 1994 -- June 10, 2016) was an American singer and YouTuber.  In 2009, she began posting covers of popular songs onto YouTube.  After releasing her debut EP, Find Me (2011), her YouTube channel reached one million subscribers.  After she reached two million subscribers, she released her debut studio album, With Love (2013).  Grimmie then became a contestant on season six of The Voice, finishing in third place.  Adam Levine, her coach on the show, announced in the finale that, regardless of the results, he would sign her to his label 222 Records.  Lil Wayne also offered |
| **Ground-truth suffix** | to sign her to his label, Young Money Entertainment.  She was signed with Island Records for a short time before being dropped.  Grimmie released her second EP, |
| **Gemini-2.0-Flash** | her a contract with his label, Young Money Entertainment.  She ultimately chose to sign with Island Records and released her second EP, Side A (2016 |
| **GPT-3.5-Turbo** | to sign her to his label Young Money Entertainment.\n\nGrimmie continued to release music independently, including her second EP, Side A (2016), which was |
| **GPT-4** | to sign her to his label, Young Money Entertainment.  However, Grimmie decided to sign with Island Records.  \n\nGrimmie's music was influenced by contemporary |
| **GPT-4.1** | to sign her to his label, Young Money Entertainment, following her impressive performances on The Voice.  Despite these offers, Christina continued to pursue her music career independently, |
| **GPT-4o** | to sign her to his label, Young Money Entertainment.  Christina Grimmie's performances on The Voice garnered her significant attention and praise, further boosting her music career.\n\nFollowing |
| **GPT-5** | to sign her to his Young Money Entertainment label; however, neither deal ultimately materialized.  Following The Voice, Grimmie released her second EP, Side A |

*Table 8.* Overall Mean MAE for estimation of per-token probabilities in logit space across Pile subsets.

| Subset | Sigmoid | PETAL | Baseline |
|---|---|---|---|
| ArXiv | 3.75 | 2.67 | 15.28 |
| BookCorpus2 | 3.28 | 1.98 | 13.25 |
| Books3 | 3.52 | 2.21 | 13.49 |
| Pile-CC | 3.62 | 2.33 | 13.43 |
| EuroParl | 4.33 | 3.57 | 14.44 |
| FreeLaw | 3.95 | 2.96 | 14.92 |
| GitHub | 6.29 | 5.78 | 16.62 |
| Gutenberg | 3.41 | 2.34 | 13.60 |
| HackerNews | 3.96 | 2.93 | 13.40 |
| DM Mathematics | 5.22 | 4.30 | 13.80 |
| OpenSubtitles | 3.46 | 2.15 | 14.35 |
| OpenWebText2 | 3.59 | 2.33 | 13.55 |
| Philpapers | 3.78 | 2.66 | 13.43 |
| StackExchange | 4.40 | 3.21 | 15.90 |
| Ubuntu IRC | 4.54 | 3.65 | 14.77 |
| USPTO | 3.42 | 2.22 | 14.43 |
| Wikipedia | 3.70 | 2.49 | 14.30 |
| YoutubeSubtitles | 4.68 | 3.65 | 14.84 |

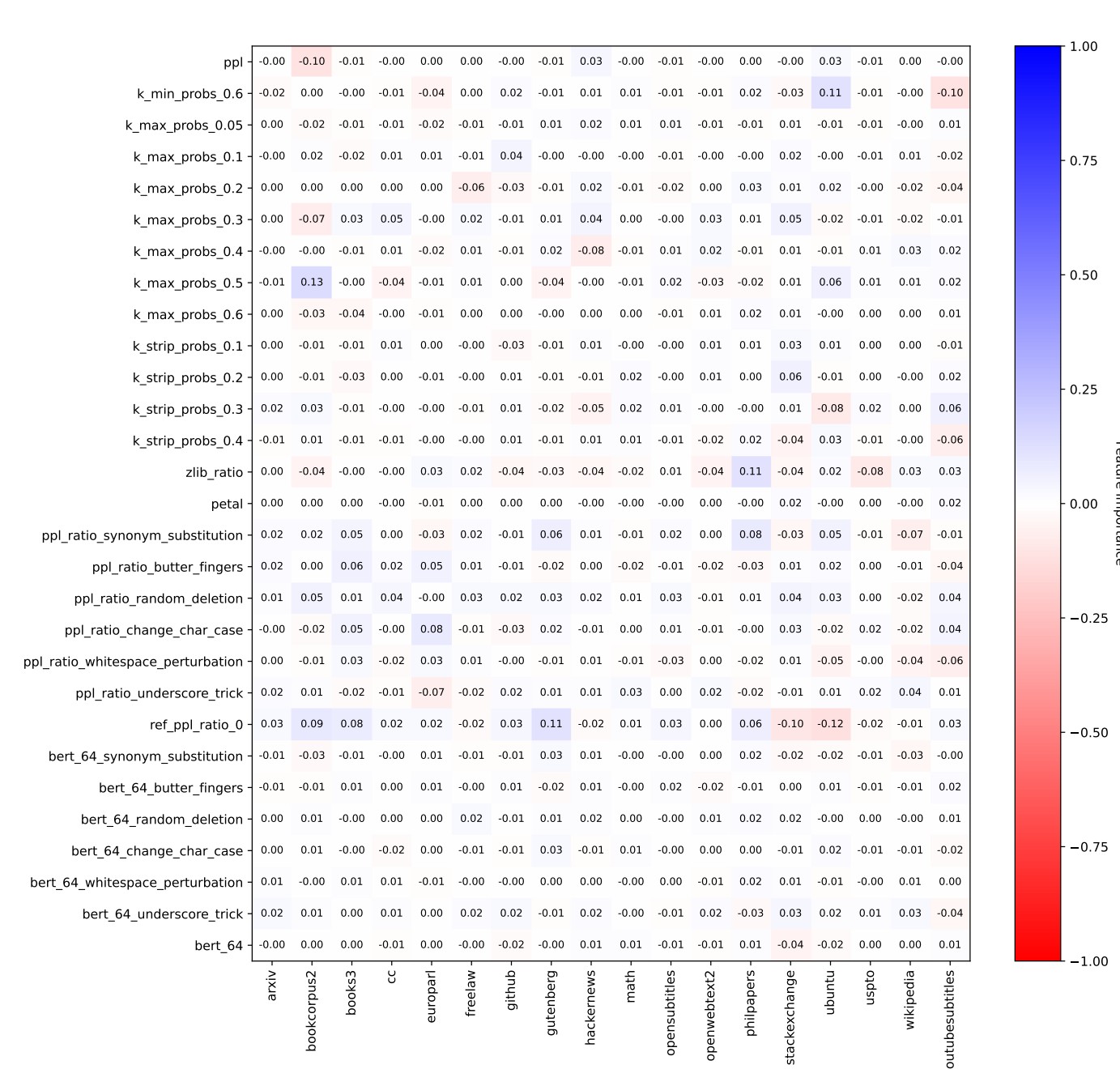

*Figure 24.* Member vs. non-member feature importance.

