# OpenReview forum: "Black-box and Adaptive Dataset Inference for Large Language Models"
_ICML.cc/2026/Conference — Submitted to ICML 2026_

### Official Review · Reviewer_nDqe · 2026-03-11

**Soundness:** 3
**Presentation:** 2
**Significance:** 2
**Originality:** 3
**Overall Recommendation:** 3
**Confidence:** 3

**Summary:**

This paper proposes a method to determine whether specific dataset was used during the training of LLMs. The method is applicable even to black-box models accessible only through an API by approximating per-token probabilities. For such black-box settings, token probabilities are estimated using a surrogate model combined with a semantic similarity measure between the surrogate model’s generated response and the actual token produced by the target model. The resulting similarity scores are then mapped to approximate token probabilities. The method is well motivated and straightforward.

**Compliance With Llm Reviewing Policy:**

Affirmed.

**Final Justification:**

The fact that membership inference performs close to random guessing does not imply that dataset inference is inherently stronger than membership inference. Instead, it mainly reflects the current difficulty of membership inference in large-scale LLMs. Dataset inference should therefore be interpreted as a relaxed objective adopted when detecting the influence of individual samples becomes infeasible.

The performance on TPR@low FPR does not appear to be significant. The two follow-up tables provided by the authors both report TPR@1% FPR, yet the results are inconsistent. In addition, the anonymous link provided by the authors contains excessive advertisements, and the figures provided in the link are difficult to access or view properly.

**Key Questions For Authors:**

1. The exposition is a little confusing. It would be helpful to define the e-value before motivating its use, or at least refer the reader to the section where it is formally introduced.

2. Why dataset inference is presented as a stronger or more meaningful threat model than membership inference?

3. Its better to provide a AUC-ROC plot to visualize TPR@Low FPR, rather than only provide a single evaluation point in a table.

**Limitations:**

See weakness and questions above. I'm welling to adjust my rating according to the authors' rebuttal.

**Strengths And Weaknesses:**

The paper introduces a novel method for estimating token probabilities in black-box settings and applying them to dataset inference attacks.

However, I do not fully understand why dataset inference is presented as a stronger or more meaningful threat model than membership inference, given that dataset inference is actually a strictly stronger notion than membership inference.

---

> ### Author Rebuttal · Authors · 2026-03-30
>
> We thank the Reviewer for their constructive feedback and finding our method novel, well motivated, and straightforward. We believe that applying the changes mentioned in the comments added a significant value to our paper. Below, we address each of your comments and questions one by one:
>
> >**W1 & Q2: why dataset inference is presented as a stronger or more meaningful threat model than membership inference?**
>
> On the technical level, [Maini et al., 2024; Duan et al., 2024] showed that membership inference attacks perform close to random guessing when applied to LLMs with billions of parameters trained on trillions of samples. In such cases, the influence of any single data point is too weak to be detected reliably. The insight of the dataset inference method is that by aggregating membership inference signals (e.g., model output probabilities, per-sample losses, etc.) over a critical number of samples, we can obtain a statistically significant signal to determine whether the suspect dataset was used in model training (even if the membership signals from individual data points are close to being non-significant).
>
> Conceptually, our goal is not to determine whether a specific sentence (e.g., from a Harry Potter book) appeared in the model’s training data, since such a sentence could easily originate from a widely available source like a Wikipedia article. Rather, we aim to assess whether the entire dataset, like a book, was included in the model’s training set, and whether the model can potentially reproduce substantial portions of that dataset, as seen in cases like Llama 3.1 70B [Cooper et al., 2025]. These issues, which exist at the dataset level rather than the single-sample level, carry significant implications for ongoing copyright litigation.
>
> >**Q1: It would be helpful to define the e-value before motivating its use, or at least refer the reader to the section where it is formally introduced.**
>
> We provided an initial definition of e-value in the introduction (please see line 87, 1st column): “an e-value is a nonnegative betting score whose expectation is at most 1 under the null (large e-values instead of small p-values indicate strong evidence)” We also added that a more detailed explanation of e-values and e-processes can be found in Section 2 on “Background and Related Work”.
>
> >**Q3: Its better to provide a AUC-ROC plot to visualize TPR@Low FPR, rather than only provide a single evaluation point in a table.**
>
> We fully agree with the reviewer that ROC curves provide a superior characterization of inference behavior across different TPR and FPR regimes. Because we share this perspective, we are pleased to point out that these evaluations were already included in the initial submission.
>
> Specifically, **Figure 11** presents the ROC curves for the Pythia model suite across the Gutenberg, PhilPapers, Wikipedia, and Ubuntu IRC subsets. Furthermore, **Figure 20** provides the corresponding ROC curves for state-of-the-art commercial black-box models (including the GPT series and Gemini 2.0 Flash) evaluated on the WikiMIA dataset.
>
> ---
> We would greatly appreciate changing the overall recommendation to "Accept" if the above responses address the Reviewer's concerns.

---

> > ### Author Rebuttal · Reviewer_nDqe · 2026-04-01
> >
> > Thanks for the authors' rebuttal. One of my concern is misunderstood. For Q3, what I'm asking for is the AUC-ROC reflecting **TPR@Low FPR** (e.g., TPR@1% FPR, TPR@5% FPR), but not the original AUC-ROC curve. You may refer to the **Figure 1 in [1]**, and provide all the TPR@Low FPR counterparts of the four subfigures in Figure 11 in an anonymous link.
> >
> > [1] Membership Inference Attacks From First Principles

---

> > > ### Author Response · Authors · 2026-04-01
> > >
> > > We thank the Reviewer for the follow-up question. We plotted the AUC-ROC curves reflecting TPR@Low FPR following Figure 1 in [1]. We provide all the TPR@Low FPR counterparts of the four subfigures in Figure 11 in the following anonymous link:
> > >
> > > https://postimg.cc/gallery/1DGfyDZ
> > >
> > > Additionally, as the Reviewer requested specifically TPR@1% FPR and TPR@5% FPR, we provide them below for the same subsets of the Pile.
> > >
> > > The following summary table shows the TPR@1%FPR performance:
> > >
> > > Dataset|Baseline|Petal|Gray-Box|BADI
> > > -|-|-|-|-
> > > Gutenberg|0.008|0.025|0.025|0.020
> > > Philpapers|0.034|0.077|0.296|0.173
> > > Wikipedia|0.000|0.006|0.079|0.091
> > > Ubuntu IRC|0.000|0.112|0.512|0.518
> > >
> > > The following summary table shows the TPR@1%FPR performance:
> > >
> > > Dataset|Baseline|Petal|Gray-Box|BADI
> > > -|-|-|-|-
> > > Gutenberg|0.058|0.060|0.090|0.108
> > > Philpapers|0.103|0.146|0.433|0.296
> > > Wikipedia|0.013|0.045|0.171|0.199
> > > Ubuntu IRC|0.000|0.216|0.601|0.592
> > >
> > > ---
> > > We would greatly appreciate changing the overall recommendation to "Accept" if the above responses address the Reviewer's concerns.

---

### Official Review · Reviewer_4cat · 2026-03-12

**Soundness:** 2
**Presentation:** 3
**Significance:** 3
**Originality:** 4
**Overall Recommendation:** 4
**Confidence:** 3

**Summary:**

This paper investigates whether a dataset was used during LLM training. Existing dataset inference methods have clear limitations. They usually require gray-box access to token-level probabilities and rely on fixed hypothesis testing based on p-values, which requires pre-determining the suspect set and a significance level. However, many commercial models are deployed through strict black-box APIs that only return generated text. It makes such methods difficult to apply. To address this issue, the authors propose Black-box and Adaptive Dataset Inference (BADI). The method first avoids direct logit access by using a sigmoid-based probability estimation strategy. It approximates token probabilities through the semantic similarity between the predicted token and the ground-truth token. It then replaces traditional fixed-sample testing with an adaptive sequential framework based on e-values. This approach enables sequential evaluation of samples, allowing the test to stop early when sufficient evidence is obtained while maintaining control of the false-positive rate.

**Compliance With Llm Reviewing Policy:**

Affirmed.

**Key Questions For Authors:**

Please refer to the weaknesses above.

**Limitations:**

yes

**Strengths And Weaknesses:**

**Strengths:**
1. Statistical Framework: The e-value–based sequential testing provides a solid statistical basis for dataset inference. Combined with ONS and Kernel MMD, it supports early stopping while keeping Type-I error under control.
2. Empirical Evaluation: Experiments are extensive, covering different model scales (Pythia-410M–12B), families (Qwen, OLMo), and commercial APIs such as GPT-4o and Gemini 2.0 Flash. The paper also includes ablations on scoring models, betting strategies, and robustness to validation contamination.
3. Clarity: The paper is clearly written and well-organized. The motivation and the shift from gray-box, fixed-sample testing to the black-box sequential framework are easy to follow.
4. Practical Value: By removing the need for logit access and fixed sample sizes, BADI makes LLM auditing more practical and data-efficient.
5. Contribution: The method offers a simple alternative to gray-box or surrogate-based approaches, combining calibration with sequential testing to achieve strong results.

**Weaknesses:**
1. Probability Estimation: The sigmoid-based probability estimation works better than PETAL in experiments, but the theoretical grounding is limited. It remains a heuristic mapping, and its performance may depend on the chosen embedding model.
2. API Cost: BADI extracts 31 features per sample, many based on perturbations. This likely requires multiple API calls. A clearer analysis of API calls or token costs to reach the stopping point compared to baselines would strengthen the efficiency claim.
3. Insufficient Baselines: Some relevant black-box DI methods are not fully evaluated. For example, DPDLLM is mentioned but not included in experiments, and CatShift only appears in the appendix. A broader comparison would make the results more convincing.

---

> ### Author Rebuttal · Authors · 2026-03-30
>
> We thank the Reviewer for the positive and constructive feedback, for finding our experiments extensive, and seeing the paper as clear and well-organized. We believe that applying the changes mentioned in the comments adds a significant value to our paper. Below, we address each question individually.
>
> >**W1-1: (...) but the theoretical grounding is limited. It remains a heuristic mapping (…)**
>
> We agree that the sigmoid-based estimation is a heuristic; however, extracting true theoretical probabilities is mathematically impossible in strict black-box settings. We found this heuristic vastly outperforms surrogate-based estimations for sequential testing due to two key advantages:
>
> 1. **Eliminating Heavy-Tailed Noise:** Surrogate models introduce heavy-tailed outlier noise due to mismatch with the suspect model (Fig. 23). By focusing purely on semantic similarity and bounding the signal to $[0, 1]$ via the sigmoid function, we eliminate these outliers, enabling the sequential test to accumulate evidence much faster.
> 2. **Direct Memorization Measurement:** Exact or highly semantically similar generation is the primary symptom of memorization. Our heuristic directly captures this strong, immediate membership signal.
>
> > **W1-2: (...) its performance may depend on the chosen embedding model.**
>
> We appreciate the insight regarding embedding model dependence. However, because our feature extraction relies on *cosine similarity*, the critical signal is relative semantic distance rather than absolute coordinates. As long as the embedding captures basic semantic equivalence, the method remains highly robust.
>
> To empirically validate this, we tested an alternative model (`stella_en_1.5B_v5`) on Pythia-12B (Ubuntu subset). Both `all-MiniLM-L6-v2` and `stella_en_1.5B_v5` achieved identical accuracy (TPR 1.0, FPR ~0.0), requiring $31.2$ and $39.3$ observations, respectively, to reach the 5% significance threshold. This confirms that the choice of the embedding model has minimal impact on pipeline performance. We included the full ablation in the revised version.
>
> >**W2: (...) A clearer analysis of API calls or token costs to reach the stopping point compared to baselines would strengthen the efficiency claim.**
>
> We thank the Reviewer for raising this practical point. To strengthen our efficiency claims, we provide a precise token and API cost breakdown. For WikiMIA-24 (128-token passages on `gpt-5`), our sequential test reaches statistical significance after just **264 samples**, utilizing two shared query types:
>
> 1. **Token-level log-probability extraction:** 127 sequential calls per sample (1 output token each). Across 264 samples, this totals **33,528 API calls** (~**2.15M input / 33.5K output tokens**).
> 2. **Suffix generation (transformations):** 7 calls per sample (original + 6 variants), each sending a 96-token prefix and receiving a 32-token output. Across 264 samples, this adds **1,848 API calls** (~**177K input / 59.1K output tokens**).
>
> **Total Cost Analysis:**
> Reaching the stopping point requires **35,376 total API calls** (~**2.32M input / 92.7K output tokens**). At standard `gpt-5` pricing ($1.25/1M input, $10.00/1M output), detecting dataset membership costs just **$3.83** (or **$1.92** via the OpenAI Batch API). For the latest `GPT-5.4` model, assuming identical observations, the cost remains highly accessible at **$7.38**.
>
> Importantly, extracting 31 features does not require 31 independent sets of queries. Because all features are derived efficiently from these two shared query types, the marginal cost of BADI's superior adaptive performance is negligible compared to standard baselines.
>
> >**W3-1: DPDLLM is mentioned but not included in experiments**
>
> We thank the Reviewer for suggesting a broader comparison. To address this, we evaluated the DPDLLM baseline on the Pythia suite (410M, 6.9B, 12B) using the Pile dataset. Sample AUC results comparing BADI and DPDLLM are provided below:
>
> |Dataset|Pythia-410M|Pythia-6.9B|Pythia-12B|
> |-|-|-|-|
> |Books3|**0.55**/0.50|**0.59**/0.52|**0.61**/0.52|
> |Github|**0.56**/0.50|**0.55**/0.50|**0.54**/0.49|
> |DM Mathematics| **0.54**/0.49 | **0.56**/0.50|**0.53**/0.49|
> |Philpapers|**0.72**/0.51|**0.74**/0.49|**0.75**/0.51|
> |YoutubeSubtitles|**0.70**/0.49|**0.69**/0.49|**0.69**/0.50|
>
> BADI offers two key advantages over DPDLLM: (1) it requires no reference model or fine-tuning, and (2) it operates sequentially online, eliminating the need for full train/test splits for classifier training. As shown above, BADI outperforms DPDLLM across all evaluated Pile subsets. Full results will be added to the camera-ready version.
>
> >**W3-2: CatShift only appears in the appendix**
>
> We relegated CatShift to the appendix because its performance was significantly worse than other baselines. This allowed us to reserve main-text space for more impactful evaluations, such as our results on real-world black-box models.
>
> ---
> We would like to kindly ask the Reviewer to further champion our work.

---

> > ### Author Rebuttal · Reviewer_4cat · 2026-04-04
> >
> > The authors have addressed my concerns, and I maintain the positive score.

---

> > > ### Author Response · Authors · 2026-04-04
> > >
> > > We thank the Reviewer for their response and for their positive feedback. We would be grateful if the Reviewer would champion our paper and increase their rating to “Accept”.

---

### Official Review · Reviewer_Y3a3 · 2026-03-12

**Soundness:** 3
**Presentation:** 2
**Significance:** 3
**Originality:** 3
**Overall Recommendation:** 4
**Confidence:** 4

**Summary:**

This paper proposes an approach to dataset inference with two key innovations:
1. Estimating token probabilities from the discrete outputs given by typical LLM APIs.
2. Using e-values to get "anytime valid" control of the type-I error rate, allowing early stopping if the evidence is conclusive.
The result is a strong and practical dataset inference attack which can be run cheaply against online models.

**Compliance With Llm Reviewing Policy:**

Affirmed.

**Final Justification:**

Following the Reply Rebuttal Comment, I raise my score to a 4.

1. I think the finding that KS can replace MMD seems like a nice finding which simplifies the method.
2. The improved ablations for sparsity, features, LASSO help show the method is well-designed.
3. Regarding the t-test, I was referring to using it as an alternative to the MMD/KS tests as a score, not to use it to replace the whole betting procedure.

**Key Questions For Authors:**

1. Both PETAL and the sigmoid based token probabilities don't really seem like "estimators" in the sense that the resulting probability distribution is essentially arbitrarily chosen. In particular the sigmoid estimator appears to give a fixed estimate that depends only on the text of the next token sampled by the model and there is no opportunity for this estimate to improve with more observations. Is this characterization correct?
2. The description of PETAL in lines 148 - 165 is quite long and may fit better in the background section.

**Limitations:**

yes

**Strengths And Weaknesses:**

Strengths:
1. The black-box feature extraction pipeline is sophisticated, leveraging both sequence and token-level features along with a simple and effective method to estimate token probabilities given only discrete token information.
2. The feature extraction uses a clever online scheme where scoring weights are learned on the fly using previously seen samples.
3. The use of e-values for testing instead of p-values seems well suited to this problem and appears to be novel. The typical concern that e-values would provide less power than a p-value based test with a sample size chosen ahead of time is allayed by the strong empirical results.

Weaknesses:

1. It is difficult to judge correctness of the test itself. The mathematical definition of the e-value is given, but it's not clear how this translates into requirements for the actual feature extraction + scoring pipeline, which is quite complex. It would be good to give the readers a high level summary of the functional requirements so it is easy to verify that the current pipeline satisfies them.
2. The system is extremely complex and the presentation makes it difficult to tell which aspects of the overall procedure are necessary for good performance. The system boasts a considerable number of innovations and features, including text perturbation, dynamic online scoring via regression, feature normalization, extreme value trimming, new features like STRIP-K% PROB, sigmoid token probability estimator, using a kernel-MMD witness for the test, and so on (there are a bunch more). Given all this, the ablation section needs to be much more comprehensive, making a clear argument that every proposed part of the pipeline is necessary for good performance. Many of the required results may already be present in the appendix, but they generally scattered and the results comparing different aspects of the pipeline aren't easily comparable. I will raise my score if this can be addressed to a satisfactory degree.

---

> ### Author Rebuttal · Authors · 2026-03-30
>
> We thank the Reviewer for the constructive feedback, for recognizing the strength of our empirical results, and for noting that our method for estimating token probabilities is both simple and effective. Below, we address each of your comments and questions individually:
>
> >**S1: (...) there is no opportunity for PETAL/Sigmoid estimate to improve with more observations. Is this characterization correct?**
>
> Yes, the characterization is correct. Our sigmoid-based approach applies a fixed mapping $P(x_t) = \sigma(sim(x_t, \hat{x}_t))$ based on the semantic similarity. It does not learn a better per-token mapping over time. Its primary purpose is to provide a numerically stable, bounded likelihood approximation that preserves directional signals for downstream features, avoiding the heavy computational overhead of querying a surrogate model (Appendix F).
>
> >**S2: The description of PETAL (...) is quite long and may fit better in the background section.**
>
> We agree with the suggestion. We moved the PETAL description to Section 2 (Background) to improve flow.
>
> >**W1: It is difficult to judge the correctness of the test itself (...) give the readers a high-level summary of the functional requirements so it is easy to verify that the current pipeline satisfies them.**
>
> To clarify how our pipeline satisfies the e-value’s mathematical requirements (Sec 4), we summarize our two mandatory components of the sequential testing:
>
> 1. **Discrepancy Metric:** Measures suspect vs. held-out distribution differences.
>
> **Requirement:** Bounded within $[-1, 1]$ (App C.1).
>
> **Implementation:** We use Kernel-MMD, squashing outputs via $\tanh$ to enforce this. App D.5 shows performance is robust compared to the naturally bounded Kolmogorov-Smirnov (KS) metric.
>
> 2. **Betting Strategy:** Determines the fraction of wealth to bet based on the observed discrepancy.
>
> **Requirement:** Prevent future "peeking", strictly bound bets to $[-\lambda_{max}, \lambda_{max}]$ (avoids ruin), and be log-optimal.
>
> **Implementation:** We process sequentially (no peeking), cap bets at $\lambda_{max}=0.5$, and use the log-optimal Online Newton Step (ONS). App D.6 tests Kelly Betting, showing consistent growth.
>
> Combined Functional Robustness Ablation (TPR):
> |Dataset|KS/Kernel-MMD|Kelly/ONS|
> |-|-|-|
> |Books3|1.0/1.0|0.8/1.0|
> |DM Math|0.04/0.06|0.04/0.06|
> |USPTO|1.0/0.9|0.4/0.9|
>
> By strictly adhering to these bounded, sequential, and log-optimal constraints, our pipeline practically and theoretically satisfies all functional requirements for anytime-valid testing.
>
> >**W2: (...) making a clear argument that every proposed part of the pipeline is necessary for good performance.**
>
> We thank the Reviewer for their feedback regarding the ablation study. Below, we clarify the strict necessity of each component through targeted experiments:
>
> * **Feature Extraction:** While L1 regularization naturally drops uninformative features, the required feature pool varies across datasets. In a new ablation, we iteratively drop features: K-Strip (Iter 1), K-Strip+Perplexity (Iter 2), and K-Strip+Perplexity+Max-K% (Iter 3). Removing features strictly degrades TPR or increases Average Stop Time (AST). Removing K-Strip drastically reduces TPR on Gutenberg. N/A indicates the test did not reach significance..
>
> |Dataset|Iter 0|Iter 1|Iter 2|Iter 3|
> |-|-|-|-|-|
> |BookCorpus2|1.0(521.6)|0.9(520.8)|0.9(595.6)|0.9(695.2)|
> |Books3|1.0(311.8)|0.9(311.2)|0.8(1127.2)|0.8(1127.2)|
> |Gutenberg|1.0(346.7)|0.9(346.4)|0.1(N/A)|0.1(N/A)|
>
> * **Online Scoring Model:** We must compress feature vectors into a testable scalar. Online LASSO adaptively aggregates features as data arrives. App A.5 shows a regularization factor of $0.01$ achieves the optimal TPR/FPR balance making online LASSO the ideal choice for both simplicity and performance.
>
> * **Trimming & Normalization:** Strictly necessary to prevent heavy-tailed outliers from destabilizing the betting process (App F). New ablations below confirm this maintains high TPR (and lower AST):
>
> |Dataset|w/ Norm & Trim|w/o Norm & Trim|
> |-|-|-|
> |BookCorpus2|1.0(521.6)|0.9(830.3)|
> |Books3 |1.0 (311.8)|0.9 (770.7)|
> |Gutenberg |1.0 (346.7)|0.9 (586.3)|
>
> * **Sequential Hypothesis Testing by Betting:** Classical p-value tests require fixed sample sizes, and any post-hoc sample addition inflates FPR. However, for sequential testing (App D.3, fig. 14), we can continuously add samples post-hoc while strictly controlling type-I error (FPR remains $\approx 0$). It is also highly robust to arbitrary non-member selection (App D.9) and severe validation pollution (up to 80% true members, App D.8).
>
> In summary: **Features** generate black-box signals $\rightarrow$ **LASSO** compresses them $\rightarrow$ **Betting** provides early-stopping guarantees. We explicitly connected these ablations in the main text of the paper.
>
> ---
> We would greatly appreciate changing the overall recommendation to "Accept" if the above responses address the Reviewer's concerns.

---

> > ### Author Rebuttal · Reviewer_Y3a3 · 2026-04-03
> >
> > Thank, you! The new results address some of my concerns, but I have some reservations about the results. First of all, the TPR results are only given to one significant figure and confidence intervals are not given for either measure. This makes it difficult to tell if the results are significant and whether the differences are meaningful.
> >
> > For LASSO, I understand the online regression is kind of "the whole point", but it would be good to show that it significantly outperforms a fixed weighing of the features. It would be even better if it outperforms the optimal fixed weighing of features (which wouldn't give a valid test, but does show that the adaptivity is crucial).
> >
> > The paper argues that the sparsity enforced by LASSO is important, but this isn't well justified empirically. I also find that it strange that in the regularization strength study, the maximum tested regularization strength is chosen. It would make more sense to keep increasing it until the results stop improving to show that the choice of regularization is correct.
> >
> > How does the Kolmogorov–Smirnov compare to kernel MMD? KS has fewer hyperparameters so if the performance is similar, why not use KS instead, or if the performance is much better with kernel MMD, it would be good to highlight this. I also do not quite understand why these scores are better than the p-value from a simple t-test, since the regression scoring seems to be explicitly trying to separate the mean scores.

---

> > > ### Author Response · Authors · 2026-04-04
> > >
> > > We thank the Reviewer for their insightful comments. Before addressing their specific questions, we provide the baseline TPR for our sequential testing pipeline (utilizing Kernel-MMD and ONS) in the table below. Establishing this baseline at the outset facilitates a clear, direct comparison for the ablation results presented in Tables 2–6:
> > >
> > > Table 1:
> > > Dataset|Results
> > > -|-
> > > BookCorpus2 (I)|$0.99\pm0.01(521.6\pm86.1)$
> > > Books3 (II)|$0.99\pm0.0(311.7\pm59.9)$
> > > Gutenberg (III)|$0.99\pm0.0(346.6\pm32.6)$
> > >
> > > Below we address each question individually:
> > >
> > > > **Q1: The TPR results are only given to one significant figure (...)**
> > >
> > > We provide the full TPR results in the Tables 2-4 below:
> > >
> > > Table 2: Combined Functional Robustness Ablation (TPR):
> > > Dataset|KS|Kelly
> > > -|-|-
> > > I|$1.0\pm0.0(167.0\pm112.6)$|$0.95\pm0.07(528.8\pm180.2)$
> > > II|$1.0\pm0.0(343.6\pm269.4)$|$0.81\pm0.16(852.1\pm224.8)$
> > > III|$1.0\pm0.0(272.6\pm159.5)$|$0.59\pm0.19(1257.5\pm230.0)$
> > >
> > > Table 3:
> > > Dataset|Iter 1|Iter 2|Iter 3
> > > -|-|-|-
> > > BookCorpus2|$0.99\pm0.01(520.8\pm83.8)$|$0.98\pm0.02(595.6\pm106.5)$|$0.97\pm (695.2\pm145.3)$
> > > Books3|$0.99\pm0.01 (311.2\pm59.8)$|$0.82\pm0.21(1127.2\pm227.1)$|$0.82\pm0.21 (1127.2\pm227.1)$
> > > Gutenberg|$0.99\pm0.00(346.4\pm31.5)$|$0.12 \pm0.18(N/A)$|$0.12\pm0.18(N/A)$
> > >
> > > Table 4:
> > > Dataset|w/o Norm & Trim
> > > -|-
> > > I|$0.95\pm0.07(830.2\pm153.3)$
> > > II|$0.92\pm0.09(770.6\pm207.9)$
> > > III|$0.98\pm0.04(586.3\pm123.5)$
> > >
> > > > **Q2: Online LASSO performance.**
> > >
> > > We compared Online LASSO against uniform and "optimal" offline fixed weightings:
> > >
> > > * Uniform Fixed Weighting (Empirically Weak): Though theoretically valid, its inability to adapt to dataset-specific memorization signals degrades performance on subsets like `github` and `freelaw`, and inflates the observations needed to reach significance.
> > > * "Optimal" Offline Weighting (Theoretically Invalid): It violates the strict *predictability* requirement of sequential testing by leaking future data. Under the null, the wealth should remain at its initial value, leading to low FPR (values close to 0.01). However, due to this violation, although we get a high TPR, this breaks Type-I error guarantees, mathematically invalidating the test and significantly inflating FPR to **0.16, 0.26, and 0.94** respectively.
> > >
> > > Thus, online adaptivity is both a theoretical and empirical necessity.
> > >
> > > Table 5:
> > > Dataset|Uniform Fixed|Optimal Fixed
> > > -|-|-
> > > I|$0.99\pm0.01(412.5\pm8.8)$|$1.0\pm0.0(249.4\pm7.8)$
> > > II|$0.99\pm0.0(303.7\pm13.4)$|$1.0\pm0.0(181.4\pm6.8)$
> > > III|$0.99\pm0.0(726.3\pm19.2)$|$1.0\pm0.0(190.2\pm7.3)$
> > >
> > > > **Q3: Sparsity.**
> > >
> > > To justify the importance of sparsity, we conducted an ablation comparing our LASSO against Ridge Regression and an OLS baseline.
> > > The degradation of OLS in Table 6 relative to the baseline confirms that regularization is necessary. Moreover, LASSO outperforms ridge, confirming that sparsity is effective.
> > >
> > > Table 6:
> > >
> > > Dataset|Ridge|OLS
> > > -|-|-
> > > I|$0.93\pm0.08(519.9\pm172.0)$|$0.64\pm0.15(888.3\pm240.7)$
> > > II|$0.91\pm0.13(624.0\pm214.4)$|$0.47\pm0.16(1218.6\pm230.1)$
> > > III|$0.86\pm0.16(733.5\pm247.9)$|$0.33\pm0.10(1447.5\pm155.4)$
> > >
> > > > **Q4: Maximum tested regularization strength**
> > >
> > > We explored higher regularization strengths during development. However, increasing the penalty beyond $0.01$ zeroes out *all* predictive features, collapsing members and non-members into a single constant and breaking the hypothesis test. Conversely, decreasing it below $0.01$ introduces noise and inflates the FPR. Thus, $0.01$ is not arbitrary, but carefully selected because it balances TPR and FPR while preserving the signal.
> > >
> > > > **Q5: KS vs Kernel-MMD**
> > >
> > > Kernel-MMD measures the distance between distribution means in an RKHS, whereas KS detects shifts by finding the maximum vertical distance between their empirical CDFs. In the paper, we experimented with both methods and presented the results for both of them in the original submission. These results (see Table 4 in the original submission) show that both methods perform roughly the same. Since, indeed as the Reviewer states, KS has fewer hyperparameters, so we are happy to suggest KS as the default choice.
> > >
> > > > **Q6: Scores**
> > >
> > > The scoring model does not only  separate means, but to learn the optimal aggregation weights needed to compress 31D feature vectors into a testable 1D scalar.
> > > As for e-values vs. p-values (t-test): T-tests mathematically require a fixed sample size; adding samples or stopping early constitutes p-hacking, which severely inflates Type-I error. E-values are fundamentally superior here because they guarantee anytime-valid Type-I error control, allowing sequential evaluation and early stopping without inflating false positives.
> > >
> > > If these responses address the Reviewer’s concerns, we would be grateful if they would raise their score.

---

### Decision · Program_Chairs · 2026-04-30

**Decision:**

Reject

**Comment:**

The paper studies the problem of dataset inference (DI), where the goal is to determine whether a given dataset contributed to an LLM's training. The main contribution is an algorithm called BADI (Black-box and Adaptive Dataset Inference), which performs DI with only black-box access to the model, i.e., access to generated tokens alone. Inference relies on sequential testing via e-values, and the authors present several experiments to illustrate the applicability of their method.

The reviewers were lukewarm overall. Reviewer Y3a3 raised issues with the presentation, noting that the paper provided insufficient detail on how the e-value formulation translates into an actual testing pipeline and that the experiments would benefit from additional ablation studies. The authors' rebuttal addressed these points through new experiments that clarified the role of each component. While these additions strengthen the paper, their  extent suggests that the manuscript would benefit from another round of revision to integrate them into the main text.

Reviewer 4cat expressed a similarly mixed view. They noted high API cost and insufficient baselines as key weaknesses. Following the rebuttal, which added a cost analysis and additional benchmarks, this reviewer marked their concerns as resolved but maintained their weak accept score.

Reviewer nDqe was the most critical. They raised concerns about the TPR@low FPR results, including inconsistencies between the tables provided in the authors' rebuttal, as well as issues with the rebuttal's anonymous link. They also pushed back on the framing of DI as a stronger threat model than membership inference.

Overall, although the paper proposes an elegant scheme for DI, it would benefit from another round of review. The additions made during the rebuttal suggest that the experimental section has room to mature further, particularly in light of the concerns raised by Reviewer nDqe. These experiments should be integrated into the main manuscript and undergo further review. The presentation could also be improved, as noted by Reviewers Y3a3 and 4cat. I am confident this paper will eventually be published -- this is a "major revision" case -- and I encourage the authors to continue to refine their work and resubmit.